# Multi-omics characterization of partial chemical reprogramming reveals evidence of cell rejuvenation

Wayne Mitchell[1], Ludger JE Goeminne[1], Alexander Tyshkovskiy[1], Sirui Zhang[1], Julie Y Chen[1], Joao A Paulo[2], Kerry A Pierce[3], Angelina H Choy[3], Clary B Clish[3], Steven P Gygi[2], Vadim N Gladyshev[1]*

[1]Division of Genetics, Department of Medicine, Brigham and Women's Hospital, Harvard Medical School, Boston, United States; [2]Department of Cell Biology, Harvard Medical School, Boston, United States; [3]Broad Institute of MIT and Harvard, Cambridge, United States

*For correspondence:
vgladyshev@rics.bwh.harvard.edu

Competing interest: The authors declare that no competing interests exist.

**Abstract** Partial reprogramming by cyclic short-term expression of Yamanaka factors holds promise for shifting cells to younger states and consequently delaying the onset of many diseases of aging. However, the delivery of transgenes and potential risk of teratoma formation present challenges for in vivo applications. Recent advances include the use of cocktails of compounds to reprogram somatic cells, but the characteristics and mechanisms of partial cellular reprogramming by chemicals remain unclear. Here, we report a multi-omics characterization of partial chemical reprogramming in fibroblasts from young and aged mice. We measured the effects of partial chemical reprogramming on the epigenome, transcriptome, proteome, phosphoproteome, and metabolome. At the transcriptome, proteome, and phosphoproteome levels, we saw widescale changes induced by this treatment, with the most notable signature being an upregulation of mitochondrial oxidative phosphorylation. Furthermore, at the metabolome level, we observed a reduction in the accumulation of aging-related metabolites. Using both transcriptomic and epigenetic clock-based analyses, we show that partial chemical reprogramming reduces the biological age of mouse fibroblasts. We demonstrate that these changes have functional impacts, as evidenced by changes in cellular respiration and mitochondrial membrane potential. Taken together, these results illuminate the potential for chemical reprogramming reagents to rejuvenate aged biological systems and warrant further investigation into adapting these approaches for in vivo age reversal.

## eLife assessment

This **important** study reports comprehensive multi-omics data on the changes induced in young and aged male mouse dermal fibroblasts after treatment with chemical reprogramming factors. The authors provide **solid** evidence to support their claim that chemical reprogramming factors induce changes consistent with a reduction of cellular 'biological' age (e.g., correlations with established aging markers in whole tissues).

## Introduction

Cellular aging is accompanied by various features, such as epigenetic changes, upregulation of inflammation, metabolic dysfunction, decreased proteostatic clearance, and increased DNA damage (*Schumacher et al., 2021*; *Hipp et al., 2019*; *Ferrucci and Fabbri, 2018*; *Amorim et al., 2022*; *Kane and Sinclair, 2019*). However, while all these hallmarks consistently show changes during aging, there

are only limited insights into their potential causative roles. Additionally, many therapies that have targeted single 'hallmarks' of aging have failed to increase lifespan or improve biological functions in vivo (*Gems and de Magalhães, 2021*; *Keshavarz et al., 2023*). An alternative way to discover therapies for targeting biological age is to develop tools to quantify it, and then screen interventions for their effect on biological age reduction. Several methods currently exist for quantifying cell aging, such as epigenetic clocks that use mean methylation levels of CpG sites that change during aging (*Horvath, 2013*), transcriptomic clocks that rely on age-associated changes in gene expression (*Choukrallah et al., 2020*; *Buckley et al., 2023*), clocks that rely on age-dependent telomere attrition (*Vaiserman and Krasnienkov, 2020*), and various proteomic, metabolomic, and glycomic clocks (*Jansen et al., 2021*; *Pearce et al., 2022*; *Johnson et al., 2020*; *Krištić et al., 2014*). The development of tools that aim to quantify biological aging have undergone significant improvements over the years. Most notably, second-generation epigenetic clocks trained to predict phenotypic measures and mortality instead of chronological age have been shown to outperform many of these aforementioned clocks in their ability to predict lifespan and healthspan (*Lu et al., 2019*).

Using various methods for biological age prediction, several intervention strategies have recently been shown to slow down and/or reverse biological aging. These include pharmacological interventions such as rapamycin and metformin (*Sharp and Strong, 2023*; *Shindyapina et al., 2022*; *Juricic et al., 2022*; *Kulkarni et al., 2020*), genetic interventions (*Petkovich et al., 2017*), heterochronic parabiosis (*Zhang et al., 2021*; *Ma et al., 2022*), and partial reprogramming by doxycycline-induced expression of Yamanaka factors (*Yang et al., 2023*; *Lu et al., 2020*; *Kriukov et al., 2022*). A natural rejuvenation process during early development has also been described (*Kerepesi et al., 2021*; *Gladyshev, 2021*). Although partial reprogramming of somatic cells has been shown to lead to improvements in biological function lost during aging (*Ocampo et al., 2016*; *Browder et al., 2022*), the effective in vivo delivery of the transgenic cassettes and precise control of the expression of Yamanaka factors present challenges. To date, only one peer-reviewed study has demonstrated lifespan increase of wild type animals upon the treatment (*Macip et al., 2024*).

Recently, administration of cocktails of small-molecule compounds have been shown to reprogram somatic cells back to pluripotency (*Guan et al., 2022*; *Hou et al., 2013*); moreover, short-term administration of two chemical cocktails (7c and 2c) has demonstrated efficacy at ameliorating several hallmarks of aging in human fibroblasts, while simultaneously preserving cellular identity (*Lucas and Paine, 2022*). Thus, partial reprogramming by short-term administration of chemicals could represent a more feasible, controllable, and adaptable method for inducing rejuvenation in vivo. However, the effects of short-term chemical reprogramming on biological age and function are currently unknown, and the mechanisms by which these chemical reprogramming cocktails act are not fully elucidated.

To address this knowledge gap, we have studied the effects of short-term partial cellular reprogramming by chemicals in both young and aged mouse fibroblasts using a multi-tiered approach. In essence, we have systematically evaluated the impact of chemical reprogramming on (i) the epigenome, transcriptome, proteome, phosphoproteome, and metabolome using various omics-based approaches, (ii) biological age with epigenetic and transcriptomic clocks, and (iii) cellular function using gene set enrichment analyses (GSEA) and measurements of cellular respiration. We demonstrate that partial chemical reprogramming induces widespread molecular changes that occur irrespective of the age of the cells. Moreover, we find significant upregulation of mitochondrial oxidative phosphorylation (OXPHOS) complexes in all treatment conditions at the transcriptome and proteome levels. Importantly, mitochondrial OXPHOS is downregulated in aging, and conversely upregulated by many lifespan-increasing interventions. Using both epigenetic and transcriptomic clocks, we show that 7c, but not 2c, reduces biological age in both young and old fibroblasts. Finally, we show that upregulation of mitochondrial OXPHOS by short-term 7c treatment strongly increases spare respiratory capacity and basal mitochondrial transmembrane potential levels. Taken together, these results imply that partial chemical reprogramming by 7c can rejuvenate several aspects of cellular aging, and that regulation of mitochondrial OXPHOS may play an important role in aging and rejuvenation.

## Results

### Partial chemical reprogramming increases pluripotency and mitochondrial OXPHOS activity

For this study, we freshly isolated tail and ear fibroblasts from young (4-month-old, or '4 month') and old (20-month-old, or '20 month') male C57BL/6 mice (*Figure 1A*). We only used fibroblasts that had been passaged ≤ 4 times to ensure that they maintain a physiologically relevant aged phenotype in cell culture. Indeed, previous studies have shown that the epigenetic age of fibroblasts increases rapidly once they are cultured in vitro (*Sturm et al., 2019*). For our study, we used two cocktails: 7c (repsox, trans-2-phenylcyclopropylamine, DZNep, TTNPB, CHIR99021, forskolin, and valproic acid), which can reprogram somatic cells to pluripotency (*Guan et al., 2022*; *Hou et al., 2013*); and 2c (repsox, trans-2-phenylcyclopropylamine), a subset of 7c that does not negatively affect cell proliferation (*Lucas and Paine, 2022*). First, we initiated treatment for 4 days prior to assessing their effects on pluripotency by staining for alkaline phosphatase (AP) activity (*Campbell, 2014*; *Figure 1B*). While neither 7c nor 2c treatment appeared to affect fibroblast morphology, 2c treatment dramatically increased the number of cells positive for AP activity in both young and old fibroblasts, an indication of increased pluripotency. In contrast, 7c treatment had no effect on AP activity.

Next, we assessed the effect of partial chemical reprogramming (6 days treatment with 2c or 7c) on resting mitochondrial membrane potential and mitochondrial mass using TMRM fluorescence. Previous studies have established a decline of mitochondrial membrane potential with age (*Hagen et al., 1997*), that caloric restriction (a well-characterized longevity intervention) acts by increasing mitochondrial membrane potential (*Berry et al., 2023a*), and that artificially increasing the mitochondrial membrane potential of *C. elegans* extends their lifespan (*Berry et al., 2023b*). Thus, characterizing the effect of partial chemical reprogramming on basal mitochondrial membrane potential may be a reporter of its efficacy in rejuvenating cells. We observed a strong increase in normalized TMRM fluorescence upon 2c and 7c treatment, and this effect was mirrored across young and old fibroblasts (*Figure 1C*). As a control, we pre-treated fibroblasts with the mitochondrial uncoupler CCCP and observed the expected decrease in TMRM signal relative to fibroblasts treated with vehicle. Thus, we concluded that partial chemical reprogramming strongly impacts mitochondrial bioenergetics during basal conditions.

Given these results, we further wanted to establish the effect of partial chemical reprogramming on mitochondrial function. To this end, we performed Seahorse experiments on fibroblasts treated with 7c and 2c (*Figure 1D*) using the Mito Stress Test protocol (*Divakaruni et al., 2014*). In order to account for the effects of these cocktails on cell proliferation, we stained and counted the cells with LCS1 immediately after the run finished, and normalized the oxygen consumption rates (OCR) to OCR per 10,000 cells. From the raw traces, we observed similar responses between young and old fibroblasts. While 2c and 7c treatment only had minor effects on basal oxygen consumption rates, 7c dramatically increased both proton leak (mitochondrial oxygen consumption with inhibited ATP synthase minus non-mitochondrial respiration) and spare respiratory capacity (uncoupled minus basal respiration). Furthermore, these results were corroborated by the observed increase in mitochondrial mass and magnitude of the mitochondrial membrane potential in response to 7c treatment. Taken together, we concluded that partial chemical reprogramming, particularly with 7c, strongly increases the activity and/or abundance of mitochondrial OXPHOS complexes.

### Partial chemical reprogramming ameliorates aging-related changes in gene expression and reduces splicing damage

To investigate the effects of partial chemical reprogramming on gene expression, and in particular on the expression of mitochondrial OXPHOS genes, we performed bulk RNA-seq (*Figure 2*) on young and old fibroblasts treated for 6 days with 7c and 2c. By principal component analysis (PCA; *Figure 2A*), we observed that fibroblasts treated with 7c are strongly separated from control and 2c treatment by principal component 1, whereas control and 2c-treated fibroblasts are separated by principal component 2. Moreover, when we assessed the association of gene expression changes caused by fibroblast age (*Figure 2—figure supplement 1A*, left panel) with previously established signatures of OSKM reprogramming and aging (*Kriukov et al., 2022*; *Tyshkovskiy et al., 2023*), we saw positive associations with multiple tissue-specific (kidney, liver) and multi-tissue (mouse and rat)

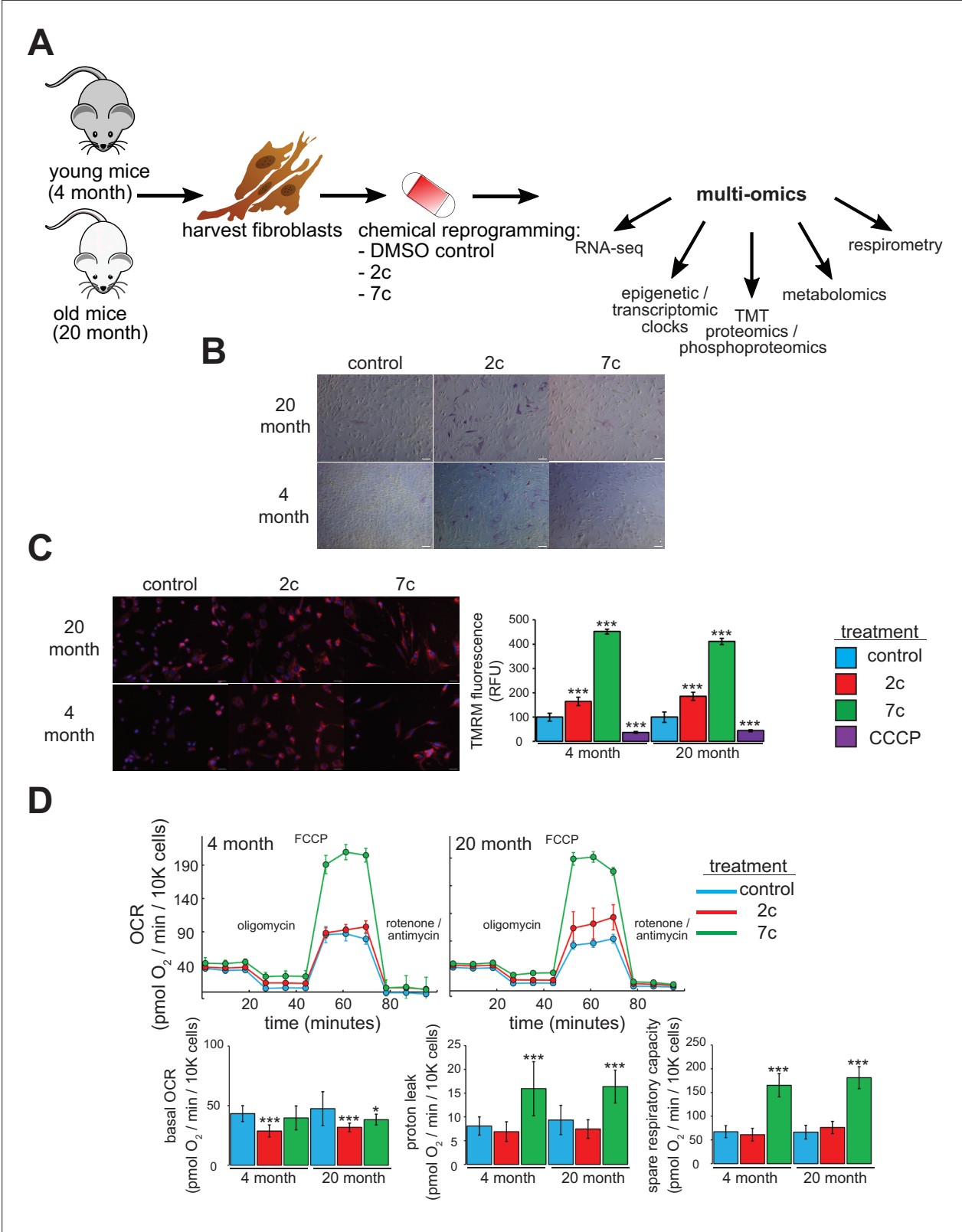

**Figure 1.** Functional effects of partial chemical reprogramming. (**A**) Overview of the study. Tail and ear fibroblasts were isolated from young (4-month-old) and old (20-month-old) male C57BL/6 mice, and cryo-stocks were prepared once reaching ~80–90% confluency (passage number P1). All fibroblasts used in this study were ≤ P4. These cells were subjected to partial chemical reprogramming followed by the indicated analyses. (**B**) AP staining. Young and old fibroblasts were treated with 2c, 7c, or DMSO for 4 days, followed by visualization of cells positive for alkaline phosphatase activity with the

*Figure 1 continued on next page*

*Figure 1 continued*

StemAb Alkaline Phosphatase Staining Kit II (4X objective). Scale bars: 126.2 µm (100 pixels). (**C**) TMRM staining. Following treatment for 6 days with 2c, 7c, or DMSO, fibroblasts were stained with 250 nM TMRM and 10 µg/ml Hoechst 33342 for 20 min at 37 °C, 5% $CO_2$, and 3% $O_2$. TMRM fluorescence intensity was normalized to the number of nuclei per field and quantified across four to five images from random fields for each independent biological replicate (n=3). For CCCP treatment, cells were treated with 50 µM CCCP in DMSO for 15 min prior to TMRM staining. Error bars represent means ± standard deviations, and data were quantified based on percent change from control-treated fibroblasts. Scale bars: 31.1 µm (100 pixels). Statistical significance was determined by one-way ANOVA and Tukey's post-hoc analysis. *$p < 0.05$, **$p < 0.01$, ***$p < 0.001$. (**D**) Effects on oxygen consumption. Top: representative raw traces of oxygen consumption of cells subjected to the Mito Stress Test protocol (basal, followed by 1 µM oligomycin a, 5 µM FCCP, and 1 µM rotenone / antimycin a) following 6 days of partial chemical reprogramming. Error bars represent means ± standard deviations from three technical replicates per treatment. Bottom: quantified oxygen consumption rates across four independent biological replicates (n=4, error bars represent means ± standard deviations). Statistical significance was determined by one-way ANOVA and Tukey's post-hoc analysis. *$p < 0.05$, **$p < 0.01$, ***$p < 0.001$.

The online version of this article includes the following source data for figure 1:

**Source data 1.** Normalized change in TMRM fluorescence.

**Source data 2.** Oxygen consumption rates.

signatures of aging, and negative associations with the mouse signature of OSKM-induced iPSCs. Thus, at the level of gene expression, the aged phenotype of the 20-month-old fibroblasts was preserved in vitro. However, the number of differentially expressed genes with 2c and 7c treatment (*Figure 2B*) far exceeded the number of differentially expressed genes modified by age (*Figure 2— figure supplement 1A*, right panel). Accordingly, changes induced by 2c or 7c were highly correlated across tested age groups (Spearman correlation coefficient > 0.8) (*Figure 2—figure supplement 1B*). We first quantified the effect of partial chemical reprogramming on the expression of pluripotency markers (*Shi et al., 2022*; *Sevilla et al., 2021*; *Lowry et al., 2008*) and the transcription factors *Klf4* and *Myc* (*c-Myc*) (*Figure 2C*). *Bend4*, *Esrrb*, and *Klf4* were all upregulated following partial chemical reprogramming, and this occurred irrespective of the age of the fibroblasts. However, 2c treatment also upregulated expression of *Myc* and *Sox2*, whereas 7c treatment did not. Furthermore, 7c somewhat paradoxically reduced expression of *Nanog* and *Myc*. Thus, 2c treatment appeared to have a more pronounced pro-pluripotency effect, which was supported by the increase in AP-positive cells following partial chemical reprogramming. Moreover, 7c had differing effects on pluripotency despite 2c being a subset of this cocktail. Thus, short-term partial chemical reprogramming with 7c appeared to show a different mechanism than normal OSKM cellular reprogramming (*Takahashi and Yamanaka, 2006*), whereas 2c in contrast appeared to share more similarities with OSKM reprogramming.

Since the accumulation of numerous types of cellular damage is thought to be causal to the aging process (*Gladyshev et al., 2021*), we next wanted to see if partial chemical reprogramming could reverse the accumulation of damaged splice variants. To this end, we characterized the alternative splicing events as a function of age (*Figure 2—figure supplement 2A*) and quantified the proportion of splicing events that may affect protein function (*Figure 2D*). For 7c-treated fibroblasts, we observed a significant lowering of splicing-related protein damage. Furthermore, we noticed a marginally significant elevation of splicing-related damage with age, whereas the relative proportions of alternative splicing events remained unchanged. To further characterize the effect of 7c treatment on alternative splicing, we assessed the distribution of ΔPercent spliced-in (ΔPsi) values of differentially spliced (|ΔPsi| > 0.1 and p-value < 0.05) genes for old (*Figure 2—figure supplement 2B*) and young (*Figure 2— figure supplement 2C*) fibroblasts, separated by the type of alternative splicing. While the number of differentially spliced events with positive and negative ΔPsi values for exon skipping was roughly equivalent, there was an excess of events with negative ΔPsi values for intron retention across both age groups. Thus, 7c treatment appeared to reduce splicing damage by decreasing the number of retained introns with functional consequences.

To further characterize gene expression changes induced by 2c and 7c, we also assessed their association with the signatures of OSKM reprogramming and aging (*Kriukov et al., 2022*; *Tyshkovskiy et al., 2023*). Consistent with previous results, we observed significant positive association between signatures of OSKM reprogramming, both in mice and humans, and gene expression changes induced by 2c treatment in fibroblasts from both age groups (*Figure 2E*). In contrast, transcriptomic profiles of fibroblasts treated with 7c did not exhibit consistent positive association with markers of OSKM-induced iPSCs. Besides, gene expression changes induced by both 2c and 7c were negatively

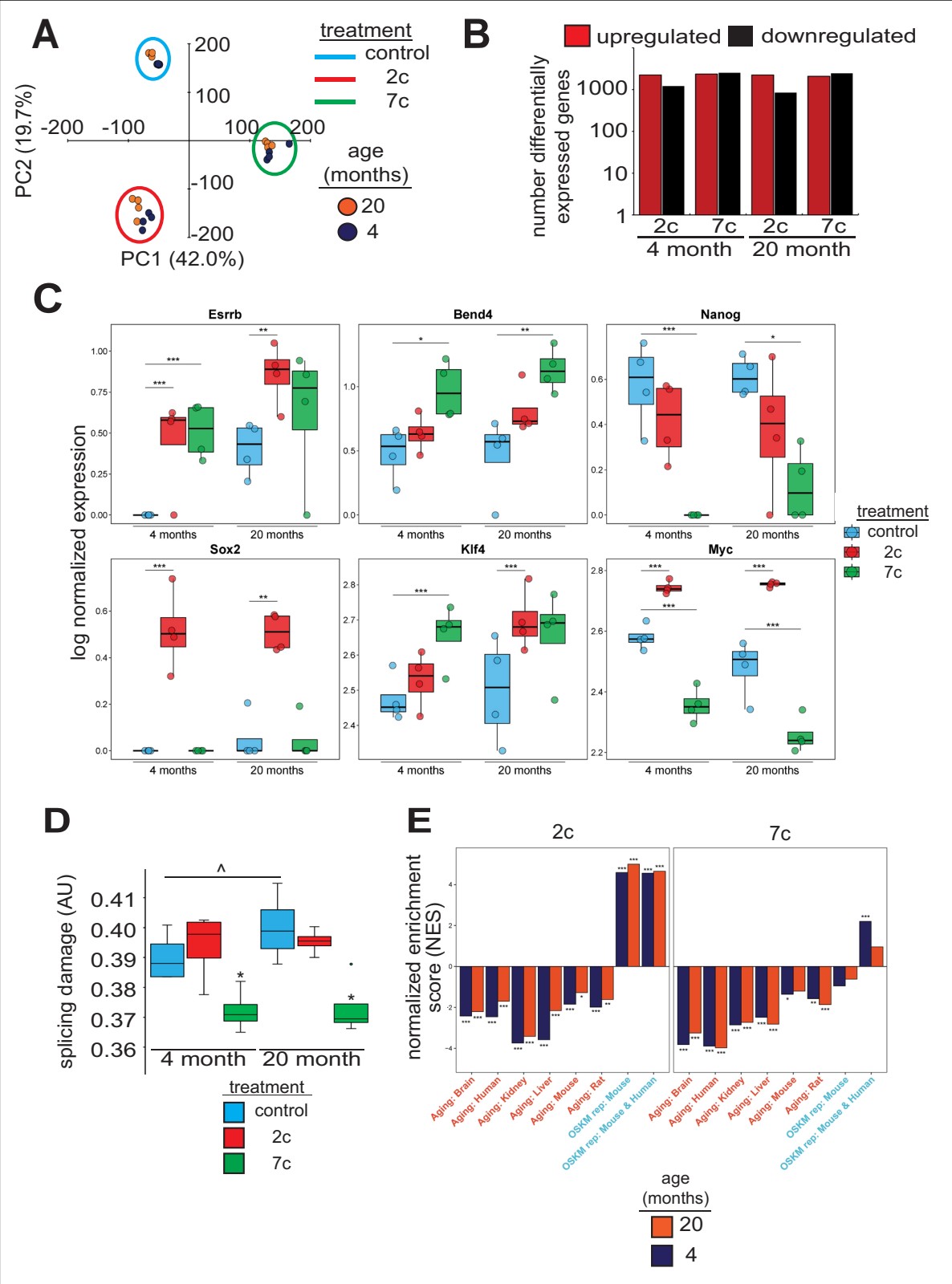

**Figure 2.** Effect of partial chemical reprogramming on gene expression. (**A**) PCA of bulk RNA-seq samples. Fibroblasts were treated with 2c, 7c, or control for 6 days followed by RNA-seq analyses. (**B**) Differentially expressed genes. Differentially expressed genes were determined using edgeR and considered statistically significant at a Benjamini-Hochberg false discovery rate (FDR) cut-off < 0.05. (**C**) Expression of pluripotency markers. Effect of partial chemical reprogramming on the normalized expression of pluripotency markers and *Klf4* and *Myc* (c-*Myc*). Statistical inference was performed

*Figure 2 continued on next page*

*Figure 2 continued*

with edgeR. *p.adjusted < 0.05, **p.adjusted < 0.01, ***p.adjusted < 0.001. (**D**) Splicing damage. Splicing damage was determined by the proportion of alternative-splicing events that may disrupt protein function. Statistical significance was determined by one-way ANOVA and Tukey's post-hoc analysis. ^p < 0.1, *p < 0.05. (**E**) Association of gene expression changes induced by chemical reprogramming with signatures of aging and OSKM reprogramming. Signatures of aging are labeled in red, whereas signatures of OSKM reprogramming are labeled in cyan. NES: normalized enrichment score. Statistical inference was done with the fgsea R package. *FDR < 0.05, **FDR < 0.01, ***FDR < 0.001.

The online version of this article includes the following source data and figure supplement(s) for figure 2:

**Source data 1.** Expression of pluripotency markers.

**Source data 2.** Splicing damage values.

**Figure supplement 1.** Effect of partial chemical reprogramming on aging-related changes in fibroblast gene and protein expression.

**Figure supplement 2.** Characterization of alternative splicing events.

associated with signatures of aging corresponding to different tissues and mammalian species, suggesting that partial chemical reprogramming may indeed counteract molecular features of aging.

## Partial chemical reprogramming increases the abundance of mitochondrial OXPHOS complexes

To elucidate the effects of partial chemical reprogramming on cellular function, we performed multiplexed proteomics (*Figure 3*) using isobaric tandem mass tags (*Li et al., 2021*). PCA based on the normalized protein abundances (*Figure 3A*) showed strong separation of 7c (principal component 1) and 2c (principal component 2) treatments from control. This effect was further visualized by heatmap (*Figure 3B*), which demonstrated that 7c treatment had a much more profound effect than 2c. Protein abundance and gene expression changes induced by 2c or 7c treatments were well correlated with each other (*Figure 3C*, *Figure 3—figure supplement 1A*), with Spearman correlation coefficients > 0.7. Furthermore, genes differentially expressed at both the mRNA and proteins levels (*Figure 3—figure supplement 1A*) following partial chemical reprogramming were generally conserved across the age groups and treatments (*Figure 3—figure supplement 1B*). Finally, by Spearman correlation analyses of proteomics samples (*Figure 3—figure supplement 1C*), we observed clustering by both treatment type and age group. Thus, we concluded that 7c treatment had a profound effect on the cellular proteome, and that our proteomics and RNA-seq analyses were in general agreement with each other.

To gain insight into the functional processes associated with the observed anti-aging effect of partial chemical reprogramming, we performed gene set enrichment analysis (GSEA) of the gene expression and protein abundance changes produced by 2c and 7c, along with signatures of aging and OSKM reprogramming (*Figure 3D*). In aging signatures, we observed a significant upregulation of pathways involved in inflammation, apoptosis, and p53 signaling, accompanied by a downregulation of genes related to energy metabolism and DNA repair. Furthermore, for aging in the fibroblast system used in this study, we also observed an upregulation of apoptosis, inflammation, and interferon signaling, thus further validating these fibroblasts isolated from aged mice as an appropriate system to study aging in vitro. When we examined all pathways affected by age at the protein level (*Figure 2—figure supplement 1C*, left panel), we further identified several other upregulated pathways classically associated with aging, including Mtorc1 and Kras signaling, and hypoxia.

In contrast, OSKM reprogramming in both human and mouse models counteracted most of these age-related changes. Interestingly, treatment with 2c and 7c also produced multiple anti-aging effects, particularly at the protein level, including upregulation of OXPHOS, TCA cycle, fatty acid metabolism, and mitochondrial translation, and downregulation of interferon signaling. Surprisingly, genes and proteins associated with the p53 pathway were upregulated by both cocktails. Partial chemical reprogramming also affected the abundance of proteins involved in cellular differentiation and extracellular matrix organization, resembling the effect of OSKM-induced iPSCs. Many of the pathways upregulated by fibroblast age at the protein level were counteracted by treatment with 7c and 2c, with Mtorc1 signaling being a notable exception in that it appeared to be even more strongly activated by 7c treatment in old fibroblasts (*Figure 2—figure supplement 1C*, center and right panels). Finally, 7c treatment resulted in a significantly lower concentration of proteins associated with PI3K/Akt signaling, suggesting that this intervention may induce pro-longevity mechanisms. Thus, partial

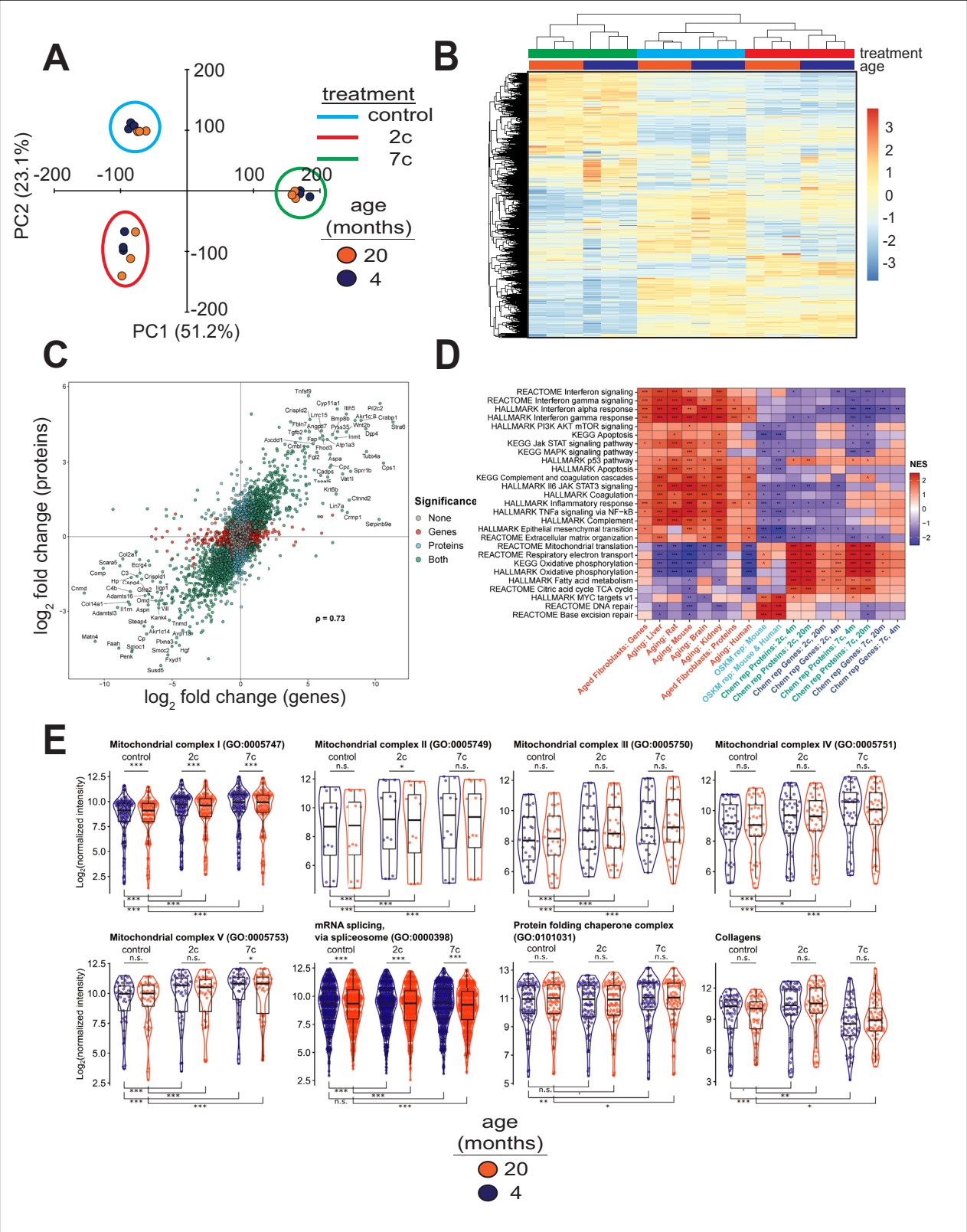

**Figure 3.** Effect of partial chemical reprogramming on protein expression. (**A**) PCA of log$_2$ normalized protein abundances. Fibroblasts were treated for 6 days with 2c, 7c, or control followed by mass spectrometry-based proteomic analyses. (**B**) Global effects of partial chemical reprogramming on the proteome. Scaled heatmap of normalized protein abundances with clustering. (**C**) Comparisons between effects on protein and gene expression. Log$_2$ fold changes of protein (vertical axis) versus mRNA (horizontal axis) for 20-month-old fibroblasts treated with 7c. Spearman correlation coefficient is

*Figure 3 continued on next page*

*Figure 3 continued*

shown in bold. Labeled are the genes that are most strongly differentially expressed after 7c treatment (adjusted p-value < 0.05) at both the mRNA and protein levels (shown in green). Also depicted are genes significantly changing only at the protein level (shown in cyan), or at the mRNA level (shown in red). Spearman correlation coefficients between protein and mRNA levels were > 0.7 for all treatment conditions (refer to *Figure 3—figure supplement 1*). (**D**) Functional GSEA. Pathways enriched by gene expression changes induced by partial chemical reprogramming (blue: 7c, green: 2c) and aging in primary fibroblasts (red) from the current study, as well as by established signatures of OSKM reprogramming (cyan) and aging (red). ^Benjamini-Hochberg FDR < 0.1, *FDR < 0.05, **FDR < 0.01, ***FDR < 0.001. (**E**) Effect of partial chemical reprogramming on the expression of protein complexes associated with aging. Protein abundances of mitochondrial OXPHOS complexes, chaperones, collagens, and the spliceosome are all affected by partial chemical reprogramming. Each datapoint represents the abundance of an individual protein. Significance was assessed with a linear regression model that included effects for gene, treatment, age group, and a treatment:age group interaction. p-values within each panel were corrected for multiple testing with the default Dunnett correction ('single-step') method in the multcomp R package (*Hothorn et al., 2008*). 'adjusted p-value < 0.1, *adjusted p-value < 0.05, **adjusted p-value < 0.01, ***adjusted p-value < 0.001, n.s. adjusted p-value ≥ 0.1.

The online version of this article includes the following source data and figure supplement(s) for figure 3:

**Source data 1.** Differential expression analysis of mRNA-seq experiments following data normalization in edgeR.

**Source data 2.** Differential expression analysis of TMT proteomics experiments following data normalization.

**Source data 3.** Gene set enrichment analysis (GSEA) of mRNA-seq experiments.

**Figure supplement 1.** Additional correlation analyses.

**Figure supplement 2.** GSEA of protein abundances.

chemical reprogramming ameliorated multiple molecular features of aging at the level of individual genes, proteins and cellular pathways, and reproduced several effects of OSKM reprogramming.

Since the anti-aging effects of partial chemical reprogramming on the proteome were more pronounced than their effects on the transcriptome, we then used GSEA on the proteomics data to find pathways affected by 7c and 2c treatments (*Figure 3—figure supplement 2A*). Many mitochondria-related processes were upregulated by partial chemical reprogramming, whereas only functions related to cell proliferation and cell cycle were downregulated. Notably, all mitochondrial OXPHOS complexes (I-V, *Figure 3E*) were upregulated for both treatments in both age groups. Proteins essential for mRNA splicing were downregulated by both age and partial chemical reprogramming, although 7c downregulated more splicing-involved proteins than 2c (*Figure 3—figure supplement 2B*). Additionally, proteins essential for RNA methylation were upregulated in both 2c- and 7c-treated fibroblasts, with more pronounced effects observed with 7c. This observation could be relevant for the reduced splicing damage after 7c treatment, as RNA methylation is known to affect mRNA splicing (*Covelo-Molares et al., 2018*). Finally, we studied the effects of partial chemical reprogramming on other proteins relevant for aging, such as chaperones (*Sóti and Csermely, 2000*), collagens (*Ewald, 2020*), and proteins implicated in the mitochondrial unfolded protein response (*Shpilka and Haynes, 2018*) (UPR^mt, *Figure 3—figure supplement 2B*). Although UPR^mt was upregulated by both 7c and 2c, chaperones were significantly upregulated only by 7c treatment, and collagen proteins were down in 7c and up in 2c. Taken together, these results suggested that partial chemical reprogramming strongly upregulated mitochondrial function, with 7c treatment having a more profound effect.

Finally, we examined Spearman correlation of changes in both mRNA and protein abundance induced by 2c and 7c with gene expression signatures of OSKM reprogramming and aging (*Figure 4*). Consistent with our previous findings, we observed a significant positive association between OSKM reprogramming features and the effects of 2c, both at the gene expression and protein abundance levels. Furthermore, the transcriptomic and proteomic changes induced by 2c and 7c were negatively correlated with multiple signatures of mammalian aging. Therefore, partial chemical reprogramming demonstrated the ability to reduce molecular signatures of aging at both the protein and mRNA levels.

## Partial chemical reprogramming phosphorylates mitochondrial proteins and activates Prkaca signaling

In addition to using multi-plexed proteomics to quantitatively ascertain overall effects on protein expression, we also evaluated the effects of partial chemical reprogramming on protein phosphorylation and kinase signaling by using TMT-based phosphoproteomics (*Figure 5*). First, we performed GSEA on the normalized relative phosphopeptide intensities. Across all treatments and age groups,

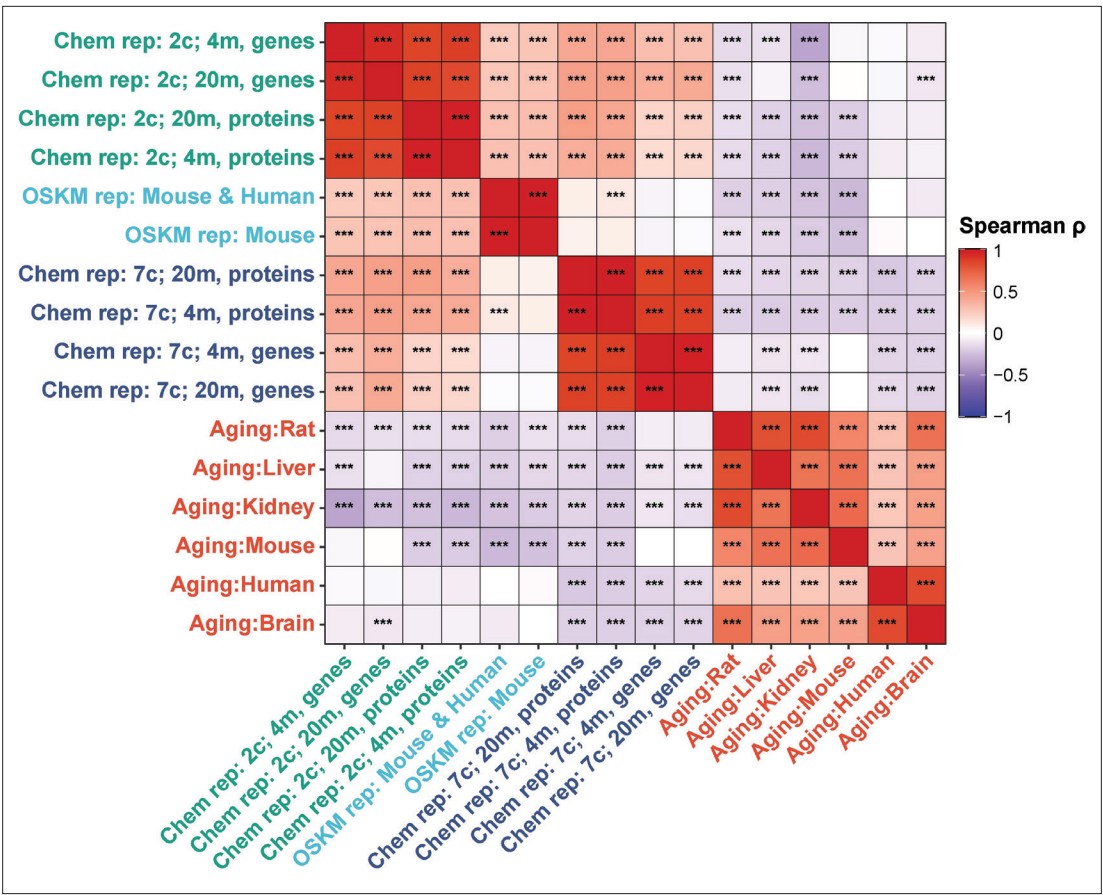

**Figure 4.** Correlation of gene expression and protein abundance changes with signatures of aging and OSKM reprogramming. Spearman correlation of gene expression and protein abundance changes induced by partial chemical reprogramming (blue: 7c, green: 2c) with the signatures of OSKM reprogramming (cyan) and aging (red). Correlation coefficients $\rho$ were calculated by the Spearman method based on the union of top 650 genes with the lowest p-value for each pair of signatures. Statistically significant pairwise correlations with Spearman $\rho > 0.1$ are labeled with asterisks. ***Benjamini-Hochberg FDR < 0.001.

we observed decreased phosphorylation of proteins involved in cell migration and proliferation, and conversely increased phosphorylation of proteins essential to fatty acid metabolism, signaling and transport of ions, and muscle development and contraction (particularly in 2c-treated fibroblasts) (*Figure 5—figure supplement 1A*). More generally, shared across 2c and 7c treatments was an upregulation in the phosphorylation of mitochondrial proteins (*Figure 5A*). In addition, for 2c-treated fibroblasts, proteins involved in lipolysis and constructing muscle were also increasingly phosphorylated. Although 7c treatment reduced splicing damage (*Figure 2E*), we observed no statistically significant changes in the phosphorylation of proteins required for mRNA splicing. Overall, we determined that 2c and 7c treatment share a common mechanism in upregulating OXPHOS activity by phosphorylating mitochondrial proteins, with 7c having a stronger effect relative to 2c. We also observed that 2c treatment activated additional pathways (lipolysis and muscle contraction) compared to 7c.

To dive deeper into the mechanisms of partial chemical reprogramming, we analyzed kinase signaling pathways affected by 2c and 7c treatments (*Figure 5B*). For all treatments and age-groups, only Prkaca signaling was significantly (FDR < 0.05) and consistently upregulated. Notably, Prkaca, the catalytic alpha subunit of the cAMP-activated protein kinase A, is crucial for regulating glucose metabolism, and upon activation, for increasing lipolysis and energy expenditure in white adipose tissue (*Turnham and Scott, 2016*). Thus, it is interesting that this kinase was activated by 7c treatment when no statistically significant effect on lipolysis was observed with this treatment. In contrast, in young fibroblasts, Akt1 and Mapk1 signaling were significantly upregulated and downregulated, respectively, upon 2c treatment. Both pathways were also strongly affected by 2c treatment in old fibroblasts but did not reach statistical significance (FDR > 0.05). Mapk1 and Akt1 have opposing effects on

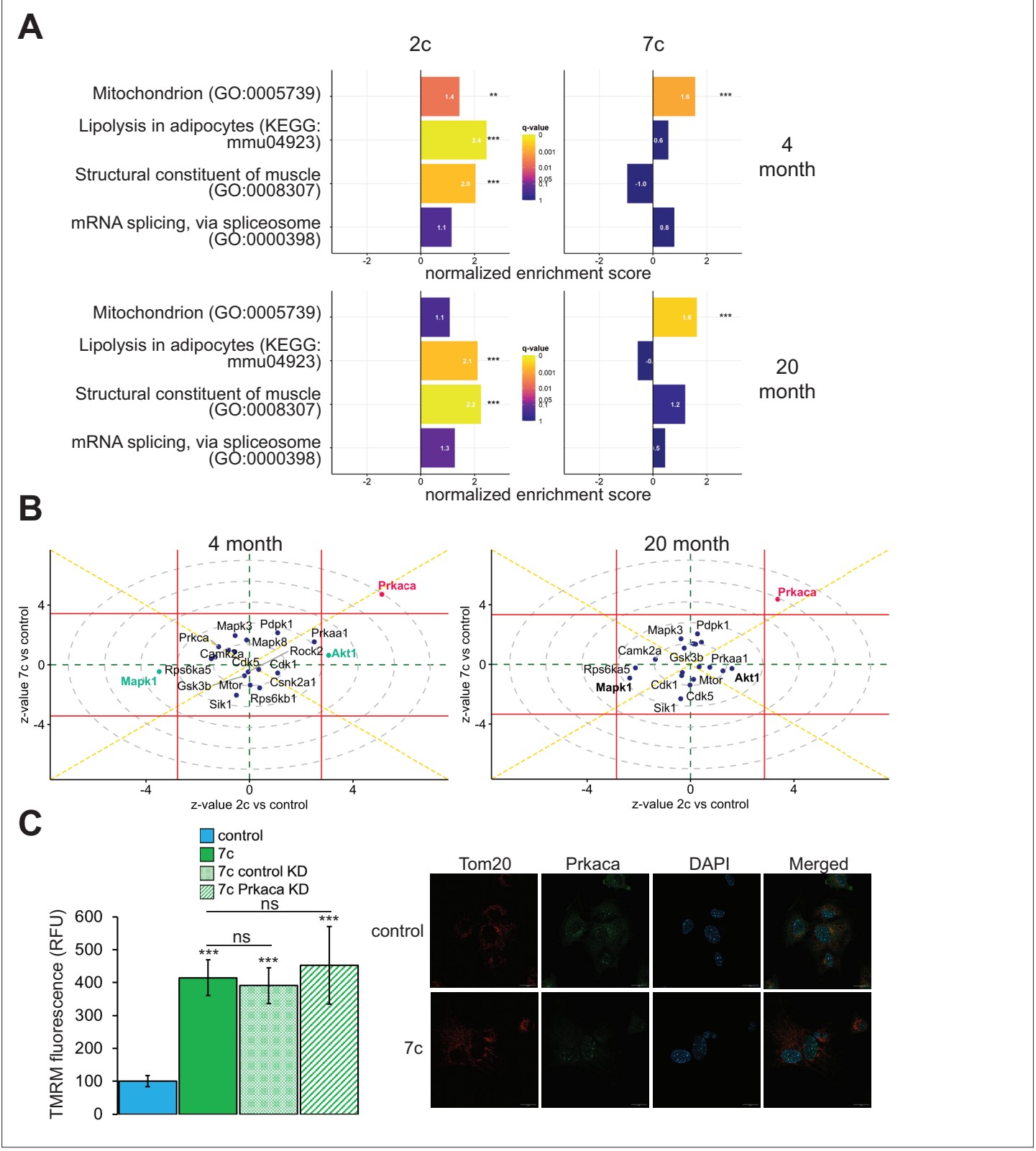

**Figure 5.** Effect of partial chemical reprogramming on the phosphoproteome. (**A**) Targeted GSEA of phosphorylation targets for four selected gene ontologies following partial chemical reprogramming. Bar lengths depict normalized enrichment scores. **Benjamini-Hochberg FDR < 0.01, ***FDR < 0.001. (**B**) Kinase signaling pathways. z-values for a kinase enrichment analysis are shown for the effects of partial reprogramming with 2c (horizontal axis) vs. 7c (vertical axis) in young (left) and old (right) fibroblasts. Signaling pathways significantly affected (Benjamini-Hochberg FDR < 0.05) by both

*Figure 5 continued on next page*

*Figure 5 continued*

7c and 2c treatment (Prkaca) are colored in red, and signaling pathways affected only by 2c in young fibroblasts (Makp1 and Akt1) are colored in green (highlighted in bold for old fibroblasts). Horizontal and vertical red lines indicate the 0.05 Benjamini-Hochberg FDR significance threshold cut-offs for the comparisons 2c vs. control and 7c vs. control, respectively. (**C**) Role of Prkaca in partial chemical reprogramming. Left: knockdown of Prkaca during 7c partial chemical reprogramming in 20-month-old fibroblasts and effects on mitochondrial membrane potential, as assessed by TMRM fluorescence (n=3 independent biological replicates). p-values were determined by one-way ANOVA and Tukey's post-hoc analysis. ns p ≥ 0.1, ***p < 0.001. Right: staining of 20-month-old fibroblasts for cellular localization of Tom20 (red) and Prkaca (green) during 7c partial chemical reprogramming. Representative images are shown, and data was collected for n=3 independent biological replicates. Scale bars: 20 μm.

The online version of this article includes the following source data and figure supplement(s) for figure 5:

**Source data 1.** Differential expression analysis of TMT phosphoproteomics experiments following data normalization.

**Source data 2.** Normalized change in TMRM fluorescence.

**Source data 3.** Uncropped blots (image files) against GAPDH and Prkaca.

**Source data 4.** Uncropped blots (labeled PDF file) against GAPDH and Prkaca.

**Figure supplement 1.** Effect of partial chemical reprogramming on cellular signaling.

regulating the expression of structural constituents of muscle (*Zhang et al., 2019*); thus, this observed influence is entirely consistent with upregulation of proteins involved in muscle contraction by 2c treatment. Therefore, we concluded that 2c and 7c share similar mechanisms of partial chemical reprogramming via activation of Prkaca signaling and mitochondrial phosphorylation. However, activation of Prkaca by 7c treatment had different downstream effects than by 2c treatment.

## Prkaca is dispensable for mediating increase in mitochondrial membrane potential following partial chemical reprogramming with 7c

Next, we wanted to verify if Prkaca signaling activated by partial chemical reprogramming was necessary to mediate its effects on the upregulation of mitochondrial OXPHOS. To this end, we knocked down Prkaca in old fibroblasts using RNA interference and assessed its impact on mitochondrial membrane potential during 7c partial chemical reprogramming (*Figure 5C*, left panel). While Prkaca was successfully knocked down to approximately 25% of its basal level as analyzed by western blot (*Figure 5—figure supplement 1B*), TMRM fluorescence of 7c-treated Prkaca knockdown fibroblasts was not statistically different from either 7c-treated control knockdown fibroblast treatments (treated with 7c only, or 7c and a non-targeted siRNA). Thus, we concluded that Prkaca signaling is either not necessary for mediating the increase in mitochondrial membrane potential caused by 7c treatment, or the amount of Prkaca remaining after knockdown is sufficient to mediate the required level of signaling. Despite this, we did observe a change in cellular distribution of Prkaca following 7c treatment (*Figure 5C*, right panel) by confocal microscopy. Instead of translocating to mitochondria, we observed that Prkaca appeared to localize to the nucleus more strongly during partial chemical reprogramming with 7c in old fibroblasts. Therefore, Prkaca could have a functional role in moderating some of the transcriptional responses to partial chemical reprogramming.

Finally, we checked the effect of partial chemical reprogramming with 7c on apoptosis in old fibroblasts using Annexin V staining and flow cytometry (*Figure 5—figure supplement 1C*). During the course of partial chemical reprogramming, we observed a steady increase in the percentage of apoptotic cells (DAPI negative, Annexin V FITC positive) that reached a maximum of approximately 20% on day 6. This percentage of apoptotic cells is similar to what has been reported with OSK/OSKM reprogramming (*Cheung et al., 2012*). Thus, these data suggest that the majority of cells successfully undergo partial chemical reprogramming by 7c treatment.

## Partial chemical reprogramming reverses the accumulation of aging-related metabolites

As the levels of endogenous cellular metabolites can have both activating and inhibitory effects on various signaling pathways, we performed an untargeted mass spectrometry-based analysis of metabolites following partial chemical reprogramming (*Figure 6*). We analyzed the abundance of polar metabolites (polar fraction taken from chloroform-methanol extraction of cell pellets) by hydrophobic interaction liquid chromatography (HILIC) (*Paynter et al., 2018; Poganik et al., 2023*). Again, to account for the effects of chemical reprogramming cocktails on cell proliferation rates, we measured

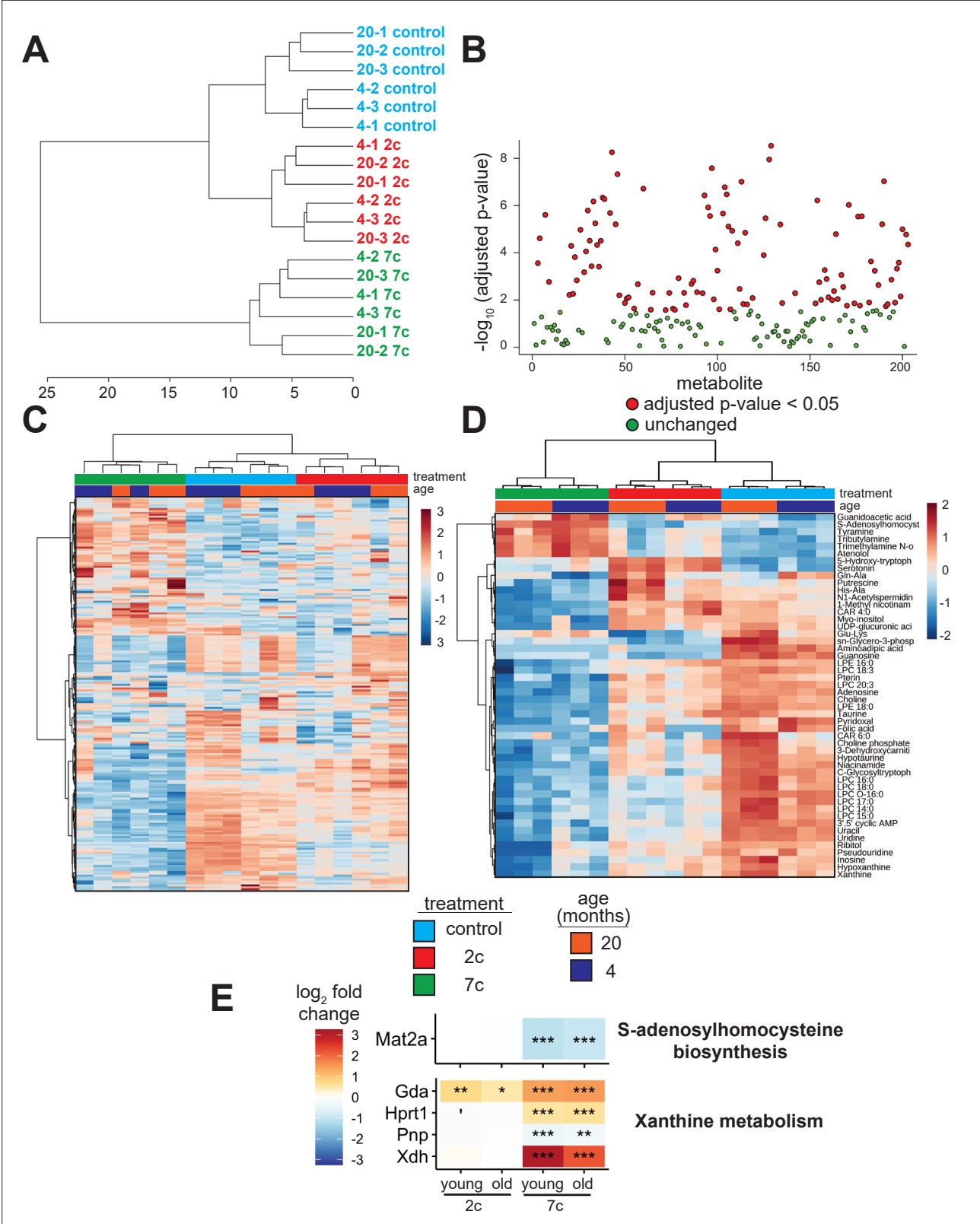

**Figure 6.** Effect of partial chemical reprogramming on the metabolome. (**A**) Hierarchical clustering (Ward's method, Euclidean distance) of polar metabolite samples. Fibroblasts were treated for 6 days with 2c, 7c, or control followed by cell scraping and collection in cold 0.9% saline solution (n=3 independent biological replicates). Polar metabolites were isolated from the frozen cell pellets by chloroform-methanol extraction, and the upper polar phase was analyzed by hydrophobic interaction liquid chromatography (HILIC) coupled to a quadrupole mass spectrometer in both positive and

*Figure 6 continued on next page*

*Figure 6 continued*

negative ionization modes (**Paynter et al., 2018**; **Poganik et al., 2023**). All subsequent analyses were performed using MetaboAnalyst 5.0 (**Xia et al., 2009**). (**B**) Metabolites affected by partial chemical reprogramming. A total of 203 metabolites were identified by HILIC-positive and -negative methods, combined. Metabolite peak areas were normalized to the amount (µg) of protein in each cell pellet (determined by BCA assay) and log-transformed. In total, abundances of 109 metabolites were significantly altered by partial chemical reprogramming (Benjamini-Hochberg FDR < 0.05, colored in red). (**C**) Global effects of partial chemical reprogramming on the metabolome. Scaled heatmap of normalized, log-transformed metabolite abundances. (**D**) Abundances of top 50 metabolites significantly altered by partial chemical reprogramming with 2c or 7c. Scaled heatmap with labeled metabolites. (**E**) Abundance of metabolite-related proteins. Abundance of proteins related to S-adenosylhomocysteine biosynthesis (top) and xanthine metabolism (bottom) following partial chemical reprogramming with 2c and 7c vs. control in young and old cells. 'Benjamini-Hochberg FDR < 0.1, *FDR < 0.05, **FDR < 0.01, ***FDR < 0.001.

The online version of this article includes the following source data and figure supplement(s) for figure 6:

**Source data 1.** HILIC-neg and HILIC-pos normalized metabolomics data.

**Figure supplement 1.** Effect of age on fibroblast metabolome.

the protein concentration in each cell pellet by BCA assay and normalized the peak area of each metabolite by the amount (µg) of protein per sample. By PCA (*Figure 6—figure supplement 1A*), we saw a clear separation of 7c treatment from control and 2c by principal component 1, whereas only young fibroblasts treated with 2c were separated from control by principal component 2. Furthermore, when we looked at just the metabolite samples from untreated fibroblasts, we observed a clear separation by fibroblast age (*Figure 6—figure supplement 1B*). All samples were separated based on their respective treatments by hierarchical clustering (*Figure 6A*), and of 203 different metabolites detected, 109 were significantly altered by the different treatment groups (*Figure 6B*).

We visualized the metabolites altered by partial chemical reprogramming by both heatmap (*Figure 6C*) and volcano plots (*Figure 6—figure supplement 1C*). These analyses revealed that the levels of many metabolites, including guanidoacetic acid, creatine, and several nucleotides, were affected by fibroblast age. Moreover, 7c treatment had a much more pronounced effect on the metabolome than 2c, which agreed with both the previous functional and multi-omics data. Since only some of the metabolites affected by fibroblast age were subsequently counteracted by 7c treatment, these data suggested that rather than targeting specific aging-related changes, 7c treatment appeared to divert cells to a different metabolic state. Analysis of the top 50 significantly altered metabolites (*Figure 6D*) revealed a cluster of 6 metabolites strongly upregulated by 7c treatment (guanidoacetic acid, S-adenosylhomocysteine (SAH), tyramine, tributylamine, trimethylamine N-oxide, and atenolol). Interestingly, SAH is known as an inhibitor of DNA methyltransferases, both by product build-up and by competitive inhibition (*Lin et al., 2020*; *Kumar et al., 2008*). Thus, the increased abundance of SAH upon 7c treatment could stem from changes in epigenetic regulation through the activity of DNA methyltransferases, especially since protein levels of Mat2a were decreased by 7c treatment (*Figure 6E*, top). Guanidoacetic acid, on the other hand, is the biosynthetic precursor of creatine and accepts a donor methyl group from S-adenosyl methionine (*McBreairty et al., 2015*) (SAM). Therefore, the effects of 7c treatment on guanidoacetic acid and SAH levels were intertwined. Conversely, many metabolites were strongly downregulated by 7c treatment, including 3',5' cyclic AMP (an activator of protein kinase A *Turnham and Scott, 2016*), xanthine (an inhibitor of phosphodiesterases that increase cyclic AMP levels *Wharton and Goz, 1979*), and various other purines, nucleosides, and nucleotides (hypoxanthine, inosine, pseudouridine, uridine, uracil, adenosine, and guanosine). In agreement with this observation, proteins involved in xanthine metabolism (*Figure 6E*, bottom) were upregulated by partial chemical reprogramming with 7c. Moreover, several purine degradation metabolites have been shown to be positively associated with aging (*Panyard et al., 2022*), suggesting that 7c treatment may reverse the accumulation of aging-related metabolites.

In the case of 2c treatment, two upregulated and related metabolites stood out: serotonin, and its biosynthetic precursor 5-hydroxy-tryptophan (5-HTP), which is decarboxylated to form serotonin (*Maffei, 2020*). Besides being an important neurotransmitter, serotonin and its precursor 5-HTP are formed downstream from tryptophan by tryptophan hydroxylase and 5-HTP decarboxylase (*Maffei, 2020*). While the abundance of tryptophan itself was not altered by partial reprogramming, the abundance of niacinamide, which is also synthesized from tryptophan (*Fukuwatari and Shibata, 2007*), was slightly reduced following 2c treatment. Lastly, for 2c, two byproducts of lysine metabolism (Glu-Lys and aminoadipic acid) were downregulated, whereas lysine levels were not significantly altered.

Since lysine is crucial for the cross-linking of collagen fibrils (*Yamauchi and Sricholpech, 2012*), and since 2c strongly upregulated the abundance of collagen proteins (*Figure 3E*), 2c treatment could be downregulating lysine catabolism. Therefore, we observed strong effects of partial reprogramming on the cellular metabolome, and these effects were generally consistent with the analyses of protein abundance and gene expression. Additionally, partial chemical reprogramming with 7c appeared to reduce the levels of several purine derivatives, which could be important for reducing aging-related accumulation of damaging metabolites.

## Partial chemical reprogramming rejuvenates both young and aged cells

To assess the rejuvenation potential of partial chemical reprogramming, we analyzed the effects of 7c and 2c treatment on fibroblast age (*Figure 7*) using mouse multi-tissue transcriptomic clocks (*Figure 7A*). As tools to measure age, transcriptomic clocks have successfully reported the effects of interventions known to affect lifespan and rate of biological aging, and appear to be sensitive to transient changes in gene expression that may occur upon short-term treatment with drugs (*Choukrallah et al., 2020*; *Buckley et al., 2023*) and OSKM reprogramming (*Kriukov et al., 2022*). Application of transcriptomic clocks developed in our lab to the bulk RNA-seq dataset (presented in *Figure 2*) revealed a statistically significant decrease of both predicted chronological and biological age with short-term 7c treatment in young and old fibroblasts; however, only the chronological clock reported a significant, but less prominent, reduction of transcriptomic age following 2c treatment. Additionally, the chronological transcriptomic clock revealed that the 20-month-old untreated primary fibroblasts were transcriptionally older than 4-month-old fibroblasts, which suggested that the fibroblast system used in this study demonstrated age-related changes in gene expression shared across multiple mouse organs. However, when the lifespan-adjusted biological clock was applied, no statistical significance was observed. Thus, we determined that the 20-month-old fibroblasts used in our study did not exhibit substantial changes in gene expression associated with decreased healthspan, while 7c had substantial effects on both general and mortality-associated age-related transcriptomic changes.

We also measured DNA methylation (DNAm) by microarrays following partial chemical reprogramming, and analyzed the results using relevant epigenetic clocks (*Mozhui et al., 2022*; *Figure 7B*). As a positive control, we measured the epigenetic ages (DNAmAge) of young and old fibroblasts reprogrammed by OSKM expression (*Figure 7—figure supplement 1*). We found a consistent lowering of DNAmAge in response to 7c treatment, whereas the effect of 2c treatment was weaker and more variable, depending on the clock used. As expected, fibroblasts isolated from old mice were epigenetically older than fibroblasts taken from young mice, which further confirmed that the fibroblasts maintained their aged phenotype in cell culture. Finally, fully-reprogrammed fibroblasts (iPSCs) demonstrated an epigenetic age of approximately 0, which was consistent with previous reports (*Horvath, 2013*). Therefore, we concluded that partial chemical reprogramming, particularly by 7c treatment, can rejuvenate both young and aged cells.

When we further analyzed the effects of fibroblast age on the epigenome, we observed a significant number of CpG sites that became differentially methylated with age (*Figure 7—figure supplement 2A*). While these aging-related differences did not impact mean DNAm levels (*Figure 7—figure supplement 2B*), we did observe an enrichment of hypomethylated CpG sites for active enhancer regions in 20-month-old fibroblasts (*Figure 7—figure supplement 2C*). When we compared these results with the effects of partial chemical reprogramming, we observed by PCA that the samples were much more strongly separated by treatment condition rather than fibroblast age (*Figure 7—figure supplement 3A*). OSKM- and 7c-reprogrammed cells were clearly separated from control samples, whereas 2c-reprogrammed cells were not. This was made further evident by the fact that 7c treatment caused a greater number of CpG sites to become significantly differentially methylated than 2c treatment (*Figure 7—figure supplement 3B*). Furthermore, rather than targeting specific CpG sites that changed with fibroblast age, 7c treatment induced global hypomethylation (*Figure 7—figure supplement 3C*) that consequently caused the reduction in cell epigenetic age. Moreover, hypomethylated CpG sites after 7c treatment were enriched for transcription and exon regions and conversely, depleted at heterochromatin and promoter regions (*Figure 7—figure supplement 3D*). This signature was shared in both young and old fibroblasts, suggesting that 7c treatment appeared to push cells to a distinct epigenetic landscape independent of their initial state.

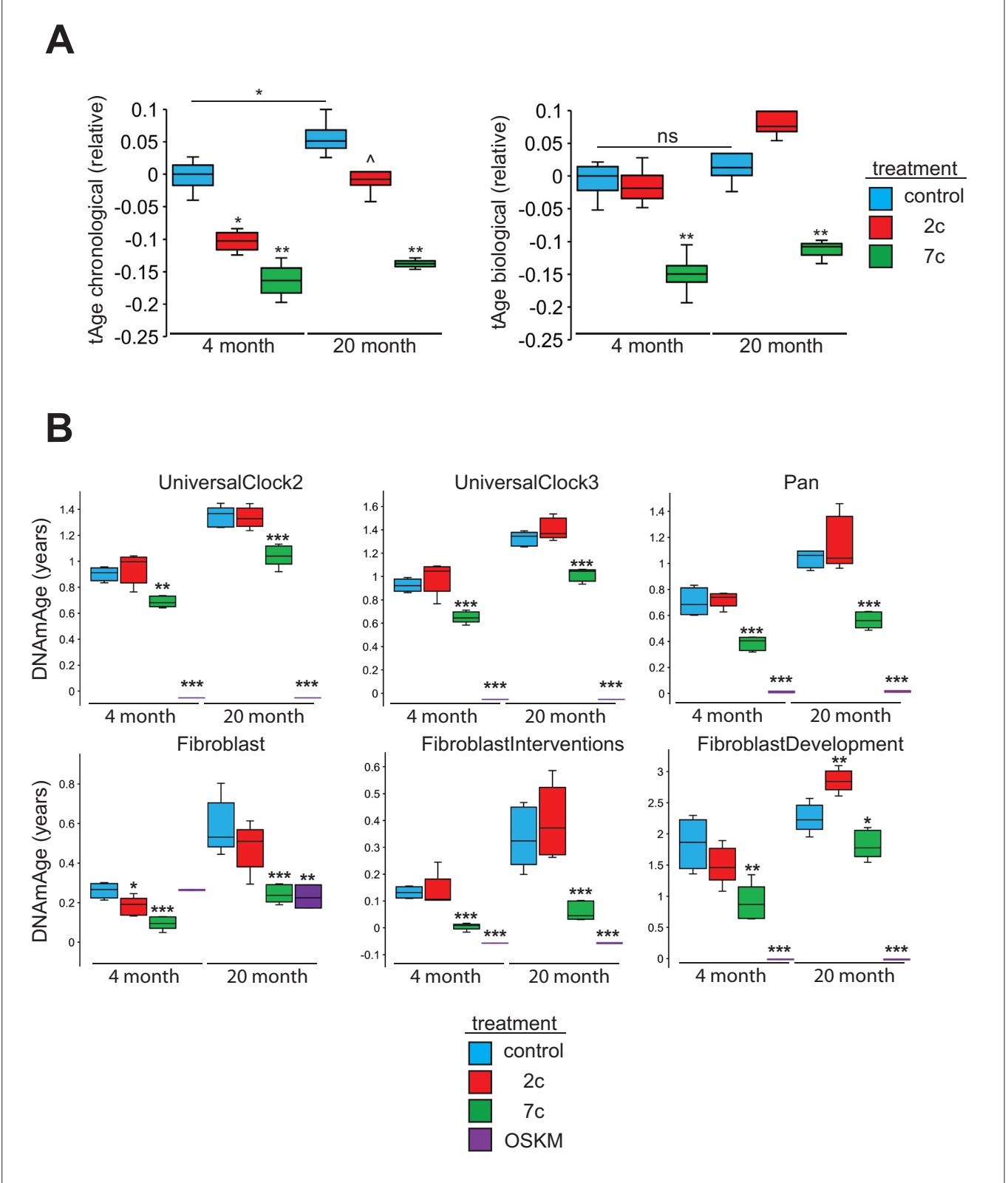

**Figure 7.** Effects of partial chemical reprogramming on biological age. (**A**) Effects of partial chemical reprogramming on transcriptomic age (tAge). tAge was assessed by using mouse multi-tissue transcriptomic clocks to analyze the bulk RNA-seq data presented in *Figure 2* (n=4 independent biological replicates). p-values were determined by one-way ANOVA and Tukey's post-hoc analysis. ns $p \geq 0.1$, ^$p < 0.1$, *$p < 0.05$, **$p < 0.01$. (**B**) Effects of partial chemical reprogramming on epigenetic age (DNAmAge). Levels of mean DNA methylation (DNAm) was assessed by DNAm microarray on the Horvath

*Figure 7 continued on next page*

*Figure 7 continued*

mammal 320k chip (n=5 independent biological replicates). p-values were determined by one-way ANOVA and Tukey's post-hoc analysis. *p < 0.05, **p < 0.01, ***p < 0.001.

The online version of this article includes the following source data and figure supplement(s) for figure 7:

**Source data 1.** Transcriptomic age predictions.

**Source data 2.** Epigenetic age predictions.

**Figure supplement 1.** Expression of pluripotency markers following OSKM reprogramming of fibroblasts.

**Figure supplement 2.** Aging-related changes in the fibroblast epigenome.

**Figure supplement 3.** Effect of partial chemical reprogramming on the epigenome.

## Discussion

The goal of our study was to characterize the effects of partial chemical reprogramming of cells at the level of the epigenome, transcriptome, proteome and phosphoproteome, and metabolome. To do this, we used low-passage fibroblasts isolated from young (4-month-old) and old (20-month-old) C57BL/6 male mice and treated the fibroblasts with two chemical reprogramming cocktails (2c and 7c). We then analyzed the effects of partial chemical reprogramming and its relationship to aging using a suite of omics-based approaches. Finally, we combined these multi-omics assays with measures of biological age (by using aging clocks) and cellular function (by utilizing respirometry and fluorescence microscopy to determine the effects of partial chemical reprogramming on mitochondrial bioenergetics). Together, these studies evaluated the potential for partial chemical reprogramming to ameliorate aging features, and uncovered its associated mechanisms for further optimization and development of treatments that could slow and/or reverse aging in vivo.

Based on measurements of biological age using epigenetic and transcriptomic clocks, we concluded that partial chemical reprogramming with 7c was able to rejuvenate fibroblasts taken from both young and old mice. Additionally, from measures of mRNA, protein, and phosphoprotein abundances, we determined that partial chemical reprogramming causes wide-scale changes to fibroblast gene expression that are larger in magnitude than the changes that occur due to aging. Based on GSEA analyses of these multi-omics data, we uncovered that the most prominent mechanistic signature of partial chemical reprogramming is the upregulation of mitochondrial oxidative phosphorylation (OXPHOS), which further manifested as a functional increase in spare respiratory capacity and increased mitochondrial transmembrane potential. Moreover, this increase in OXPHOS activity was accompanied by a slight increase in apoptosis during the course of reprogramming, suggesting that partial chemical reprogramming might remove cells with dysfunctional mitochondria. Finally, we showed that partial chemical reprogramming can reverse molecular damage positively associated with aging, including the accumulation of aging-related metabolites and mis-spliced mRNA.

The downregulation of mitochondrial OXPHOS and translation is a known signature of aging (*Tyshkovskiy et al., 2019*; *Tyshkovskiy et al., 2023*); furthermore, several recent studies have highlighted that augmenting the mitochondrial transmembrane potential can increase longevity (*Berry et al., 2023a*; *Berry et al., 2023b*). In this light, we suggest that partial chemical reprogramming of fibroblasts induces rejuvenation, at least in part, by upregulating mitochondrial OXPHOS. Importantly, this differs significantly from OSKM reprogramming, which shows a strong glycolytic shift in response to the induction of OSKM expression (*Nishimura et al., 2019*). Furthermore, our results are consistent with the activation of different transcription factors by partial chemical reprogramming with 7c, which showed negligible expression of the Yamanaka factors, increased expression of two pluripotency markers (*Esrrb* and *Bend4*), and decreased expression of *Nanog*.

The results of several recent studies have illustrated the changes in mitochondrial ultrastructure that occur during aging (*Brandt et al., 2017*; *Lu et al., 2022*; *Stahon et al., 2016*), which may result from several interdependent mechanisms such as disrupted Opa1-, Drp1-, and Micos-mediated fission and fusion dynamics (*Warnsmann et al., 2022*; *Zer et al., 2022*; *Pellegrini and Scorrano, 2007*; *Sharma et al., 2019*; *Rana et al., 2017*), decreased cardiolipin content (*Semba et al., 2019*), destabilized ATP synthase dimers (*Daum et al., 2013*), and uncontrolled opening of the mitochondrial permeability transition pore (*Rottenberg and Hoek, 2021*; *Angeli et al., 2021*). Importantly, mitochondrial ultrastructure determines bioenergetic function and thus, detrimental changes to mitochondrial

ultrastructure (such as loss of tight cristae junctions and cristae curvature) can lead to bioenergetic declines during aging (*Sun et al., 2016*; *Gómez and Hagen, 2012*). These declines include increased proton leak and reactive oxygen species production (*Sun et al., 2016*; *Gómez and Hagen, 2012*; *Zhang et al., 2020*), decreased respiratory supercomplex assembly and associated spare respiratory capacity (*Gómez and Hagen, 2012*; *Novack et al., 2020*), and impaired proteostasis (*Patron et al., 2022*; *Moehle et al., 2019*). Thus, the activation of factors that reinforce proper mitochondrial ultrastructure could reverse many aging-related elements of mitochondrial dysfunction.

It should be emphasized that across most assays, the effects of partial chemical reprogramming were similar in both young and old fibroblasts. In simplest terms, this could be because the changes observed due to fibroblast aging (i.e. in terms of the number of differentially expressed genes, proteins, and metabolites) were approximately an order of magnitude lower than those observed with partial chemical reprogramming. An alternative, non-mutually exclusive explanation is that partial chemical reprogramming, particularly with 7c, pushes cells to a different metabolic state that makes them biologically younger (i.e. at the transcriptional and epigenome levels) as a byproduct. For example, with the untreated fibroblasts, we observed an increased epigenetic age in the 20-month-old fibroblasts vs. the 4-month-old fibroblasts without a change in mean DNAm. In contrast, with 7c treatment, we saw decreased epigenetic age in both young and old fibroblasts that occurred due to induced global hypomethylation. Thus, we believe that 7c-mediated partial chemical reprogramming may activate a unique set of transcription factors that produce an epigenetically-distinct cellular state, irrespective of the initial state of the epigenome and in addition to the known de-methylating effects of DZNep.

In conclusion, we have shown that partial chemical reprogramming is able to reverse many signs of aging in a mouse fibroblast model. We thus posit that partial reprogramming by chemicals may represent a viable alternative to the cyclically-induced, ectopic expression of Yamanaka factors for rejuvenating aged cells. There are several limitations of OSK/OSKM partial reprogramming, including low efficiency (*Takahashi and Yamanaka, 2006*; *Ohnishi et al., 2014*), oncogenic risk (*Ohnishi et al., 2014*), and potential toxicity due to varying cell types being reprogrammed at different rates (*Singh and Zhakupova, 2022*). In this study, we have shown that partial chemical reprogramming with 7c features a distinct mechanism from OSK/OSKM partial reprogramming; thus, partial chemical reprogramming could potentially circumvent these risks because it activates different transcriptional programs to reduce cellular biological age. Furthermore, as partial chemical reprogramming utilizes druggable small-molecule compounds, dosages and delivery agents can be precisely tailored to modify their pharmacokinetics and tissue distributions, and to minimize their cellular toxicity and potential off-target effects. Further testing however, particularly in animal models of aging, is needed to fully elucidate the effectiveness of partial chemical reprogramming in vivo, and to establish what advantages, if any, it may have over OSK/OSKM partial reprogramming. In addition, it is important to note that this study was performed using only one cell type isolated from inbred male mice; different cell types experience varying changes during aging (*Kimmel et al., 2019*), and the effects of aging and lifespan-extending interventions are generally sex-dimorphic (*Sampathkumar et al., 2020*). Thus, extensive validation in multiple cell types and in genetically-diverse male and female mice is necessary to determine if partial chemical reprogramming can ameliorate several relevant dimensions of mammalian aging. In addition, a more thorough characterization of the effect of partial chemical reprogramming on mitochondrial dynamics and function, such as OXPHOS supercomplex assembly, cristae ultrastructure, and protein import, could shed greater light on the importance of various mitochondrial processes in the aging process. Finally, elucidating the molecular targets of chemical reprogramming compounds, whether they may be transcription factors, regulators of the epigenome, and/or various kinases and cellular receptors, could further contribute to our understanding of aging and how to best reverse it.

## Materials and methods
### Materials and animals
Young (4-month-old) and old (20-month-old) male C57BL/6 mice were acquired from the National Institute on Aging (NIA) aged rodent colonies (Charles River Laboratories, Wilmington, MA) and were fed standard 5053 chow. HEK293T cells were acquired from ATCC (Manassas, VA) and were verified by

PCR to be free of mycoplasma contamination. Cell culture reagents were acquired from Thermo Fisher Scientific (Waltham, MA), reagents for isolating DNA and RNA were from Qiagen (Hilden, Germany) and Zymo Research (Irvine, CA), respectively, and general preparatory chemicals and reagents were from Sigma-Aldrich (St. Louis, MO). Chemical reprogramming reagents were obtained from the following suppliers: Sigma-Aldrich (repsox, trans-2-Phenylcyclopropylamine, DZNep), Cayman Chemical Company (TTNPB, CHIR99021), Tocris Bioscience (forskolin), and STEMCELL Technologies (valproic acid). Tetramethylrhodamine, methyl ester perchlorate (TMRM) was obtained from MedChem Express (Monmouth Junction, NJ). Antibodies were obtained from Abcam (Cambridge, United Kingdom). All experiments using mice were performed in accordance with institutional guidelines for the use of laboratory animals and were approved by the Brigham and Women's Hospital and Harvard Medical School Institutional Animal Care and Use Committees under Protocol #2016N000368.

## Isolation of mouse fibroblasts

Fibroblasts were isolated from the ears and tails of male C57BL/6 mice according to a published procedure (*Khan and Gasser, 2016*). In brief, mice were euthanized by exposure to $CO_2$ and cervical dislocation, after which ears and ~5 cm of tail were sterilized in 70% ethanol for 5 minutes. After air drying, tissues were cut into small, ~3 mm pieces and digested with a mixture of collagenase and pronase for 90 minutes at 37 °C with gentle shaking. Following enzymatic digestion, the tissues were ground in 70 µm cell strainers using a 10 ml syringe plunger, and the resulting cell suspension was centrifuged at 500xg for 5 minutes at room temperature. The cell pellet was resuspended in fresh fibroblast culture media (DMEM/F12, 10% FBS, 1 X antibiotic-antimycotic, 50 µM β-mercaptoethanol, 1 X non-essential amino acids), spun once more at 500xg, resuspended in 10 ml fibroblast culture media, plated in 10 cm tissue culture dishes, and incubated for 3 days at 37 °C, 5% $CO_2$, and 3% $O_2$. Following 3 days of incubation, media was replaced to get rid of cellular debris, and the cells were then incubated further until reaching ~80% confluency. Upon reaching the desired confluency, ear and tail fibroblasts were trypsinized and frozen at 1 million cells per cryovial in fibroblast freezing media (fibroblast culture media with 50% FBS and 10% DMSO).

## Production of lentivirus

HEK293T cells were cultured in high glucose DMEM plus sodium pyruvate, GlutaMAX, 10% FBS, and 1 X antibiotic-antimycotic, and were grown to ~70% confluency at 37 °C, 5% $CO_2$. Viral vectors VSVG and gag/pol 8.91, along with either the polycistronic OSKM cassette (*Carey et al., 2009*) (Addgene #20328) or the reverse tetracycline transactivator (*Hockemeyer et al., 2008*) (Addgene #20342), were encapsulated in PureFection nanoparticles (System Biosciences, Palo Alto, CA) and added drop-by-drop to the culturing cells. After 48 and 72 hours, the cell culture media was collected through a 0.45 um filter, spun at 3,000 RPM for 15 minutes at room temperature, and the resulting supernatant was aliquoted and stored at –80 °C.

## OSKM reprogramming of fibroblasts

Fibroblasts were seeded in six-well plates at a cell density of 100,000 cells per well in fibroblast culture media and were incubated for 24 hours at 37 °C, 5% $CO_2$, 3% $O_2$. Fibroblasts were subsequently co-transfected with 1 ml of each lentivirus (OSKM and reverse tetracycline transactivator), 8 µg/ml polybrene, and 1 ml fresh fibroblast culture media for 3–4 days. Following 3–4 days of transfection, cells were trypsinized and plated on Geltrex (Thermo Fisher, Waltham, MA) coated dishes and incubated for 48 hours. Then, the media was aspirated and replaced with ES media (fibroblast culture media with 20% FBS, 1000 U/ml mouse Leukemia Inhibitory Factor (LIF), and 2 µg/ml doxycycline) and the cells were cultured for 1–2 weeks until colonies appeared (media replaced every 3 days). Colonies were then manually picked, trypsinized, and plated in Geltrex coated 24-well plates in N2B27 (1:1 mixture of DMEM/F12: 1X N2 supplement, 7.5% BSA, 1 X non-essential amino acids, 100 µM β-mercaptoethanol, 1 X antibiotic-antimycotic, and Neurobasal: 1 X B27 supplement, 7.5% BSA, 1 X non-essential amino acids, 1 X GlutaMAX, 100 µM β-mercaptoethanol, 1 X antibiotic-antimycotic) media, plus 1 µM PD0325901, 3 µM CHIR99021, 0.5 µM Stemolecule A83-01, 1000 U/ml LIF, and 2 µg/ml doxycycline. Colonies that survived in this media were deemed to be fully reprogrammed iPSCs and were further propagated into Geltrex coated 10 cm dishes and cryofrozen in 50% N2B27 media (without inhibitors or doxycycline), 40% FBS, and 10% DMSO at a cell density of 2 million cells per vial.

## AP staining

Fibroblasts were seeded at a density of 100,000 cells per well in six-well plates, and cells were treated with the specified compounds for 4 or 6 days at 37 °C, 5% $CO_2$, 3% $O_2$ (media and treatments replenished every 3 days). Cells were washed twice with PBS-T, fixed, and stained with the Stemgent AP Staining Kit II according to the manufacturer's protocol (ReproCELL USA, Inc Beltsville, MD). Cells were then imaged under a brightfield microscope.

## Membrane potential staining

Fibroblasts were stained with 250 nM TMRM and 10 μg/ml Hoechst 33342 in fibroblast culture media for 20 minutes at 37 °C, 5% $CO_2$, and 3% $O_2$. Then, the media was aspirated, and fibroblasts were washed three times with PBS before imaging in the dark on an AXIO Observer.Z1 (Zeiss, Oberkochen, Baden-Württemberg, Germany). As a positive control, cells were pre-treated with 50 μM of carbonyl cyanide m-chlorophenylhydrazone (CCCP) for 15 minutes prior to incubation with TMRM. Images taken from random fields (4–5 images per treatment condition per biological replicate) were background-subtracted in ImageJ (*Schneider et al., 2012*), and mean fluorescence was normalized to the number of nuclei per field.

## Immunocytochemistry

Cells in 24-well plates were rinsed twice with PBS, fixed in 4% paraformaldehyde for 20 minutes at room temperature, and rinsed twice more with PBS. Then, cells were permeabilized with 0.3% Triton X-100 in PBS for 5 minutes, followed by blocking with 3% BSA for 1 hour at room temperature with gentle shaking. After this, cells were incubated with the indicated primary antibody overnight at 4 °C, rinsed in the morning with PBS for 2 cycles (10 minutes each), and incubated with fluorophore-conjugated secondary antibody for 1 hour at room temperature in the dark. Cells were then finally stained with 10 μg/ml Hoechst 333342 for 10 minutes, rinsed three times with PBS, and imaged on an AXIO Observer.Z1 inverted fluorescence microscope (Zeiss, Oberkochen, Baden-Württemberg, Germany).

## DNA methylation

DNA was isolated from cells using the DNeasy Blood and Tissue kit according to the manufacturer's instructions (Qiagen, Hilden, Germany). Concentration of DNA samples was measured using the Qubit dsDNA BR kit (Invitrogen, Waltham, MA). DNAmAge (using various epigenetic clocks) was assessed using DNA methylation data obtained on the Horvath mammal 320k array (*Arneson et al., 2022*) through the Epigenetic Clock Development Foundation (Torrance, CA). Raw methylation array data was normalized using SeSAMe (*Zhou et al., 2018*). Probes were filtered and removed based on the following criteria: missing values or a p-value of detection > 0.01 for any samples, not mappable to the mouse *mm10* reference genome and technical controls for the array.

All DNA methylation analysis was performed in the R/Bioconductor environment. Probe-wise differential methylation analysis was performed using limma using M-values, since this method has been shown to be more statistically robust for identifying significantly differentially methylated probes (DMPs) when compared to using beta values (*Du et al., 2010*). Significantly DMPs were identified using a cutoff threshold where |log2FC(M-value)| > 1 and Benjamini-Hochberg false discovery rate < 0.05. Enrichment of the significantly DMPs for specific chromatin states was assessed using a universal chromatin state annotation of the mouse genome spanning 26 cell and tissue types (*Vu and Ernst, 2023*). Briefly, we calculated odds ratios of observing significantly up- or down-regulated DMPs in each chromatin state and performed a one-sided hypergeometric test (*phyper*) to test for significance of enrichment or depletion. We report signed $\log_{10}$ p-values, adjusted using the Benjamini-Hochberg method, of either enrichment (OR >1, sign = +1) or depletion (OR ≤ 1, sign = –1) depending on the odds ratio.

## RNA-seq

Fibroblasts were seeded in six-well plates at a cell density of 100,000 cells per well in fibroblast culture media and were incubated for 6 days at 37 °C, 5% $CO_2$, 3% $O_2$ with their respective chemical treatments (media replaced once after 3 days). On the 6th day, media was aspirated, and the cells were washed twice with PBS. Then, the cells were directly lysed with 700 μl of Trizol, diluted 1:1 with

200 proof ethanol, and total RNA was isolated using a direct-zol mini prep kit (Zymo Research, Irvine, CA). RNA concentration was measured by Qubit using the RNA HS assay kit (ThermoFisher, Waltham, MA). Libraries were prepared with TruSeq Stranded mRNA LT Sample Prep Kit according to TruSeq Stranded mRNA Sample Preparation Guide, Part # 15031047 Rev. E. Libraries were quantified using the Bioanalyzer (Agilent) and sequenced with Illumina NovaSeq6000 S4 (2×150 bp) (reads trimmed to 2×100 bp) to get 20 M read depth coverage per sample. The BCL (binary base calls) files were converted into FASTQ using the Illumina package bcl2fastq. Fastq files were mapped to the *mm10* (GRCm38.p6) mouse genome, and gene counts were obtained with STAR v2.7.2b (*Dobin et al., 2013*). Statistical analyses for gene expression were performed with custom models from DEseq2 (*Love et al., 2014*) 3.13 and edgeR (*Robinson et al., 2010*) 3.34.1 in R. We filtered out genes with low number of reads, keeping only the genes with at least 5 reads in at least 20% of the samples, which resulted in 16,728 detected genes according to Entrez annotation. Filtered data was then passed to Relative Log Expression (RLE) normalization. Differential expression of genes perturbed by compounds was analyzed using edgeR. Obtained p-values were adjusted for multiple comparison with the Benjamini-Hochberg method.

## Association with gene expression signatures of aging and OSKM reprogramming

Association of gene expression log-fold changes in aged fibroblasts and induced by chemical reprogramming cocktails with established transcriptomic signatures of aging and OSKM reprogramming was examined with GSEA-based method as described previously (*Tyshkovskiy et al., 2019*). The utilized signatures of aging included tissue-specific liver, kidney, and brain signatures as well as multi-tissue signatures of the mouse, rat, and human (*Tyshkovskiy et al., 2023*). Signatures of OSKM reprogramming included genes differentially expressed during cellular reprogramming of mouse fibroblasts (Mouse), and shared transcriptomic changes during OSKM-induced reprogramming of mouse and human fibroblasts (Mouse and Human) (*Kriukov et al., 2022*). For every signature, top 500 statistically significant genes (adjusted p-value < 0.05) with the highest absolute logFC were used as gene sets for GSEA-based association analysis. Pairwise Spearman correlation for gene expression and protein abundance changes induced by chemical reprogramming as well as transcriptomic signatures of aging and OSKM reprogramming was calculated based on the union of top 650 genes with the lowest p-value for each pair of signatures.

For the identification of enriched functions affected in aged fibroblasts or by 2c or 7c treatment, we performed functional GSEA (*Subramanian et al., 2005*) on a pre-ranked list of genes or proteins based on $\log_{10}$(p-value) corrected by the sign of regulation, calculated as:

$$-log\left(pv\right) \times sgn\left(lfc\right)$$

where *pv* and *lfc* are p-value and logFC of a certain gene, respectively, obtained from edgeR output, and *sgn* is the signum function (equal to 1,–1 and 0 if value is positive, negative or equal to 0, respectively). Gene ontology: biological process (GObp), HALLMARK, KEGG and REACTOME ontologies from the Molecular Signature Database (MSigDB) were used as gene sets for GSEA. The GSEA algorithm was performed separately for each cocktail and age group via the fgsea package in R with 5000 permutations. p-values were adjusted with the Benjamini-Hochberg method. A false discovery rate cutoff of 0.1 was used to select statistically significant functions. A similar analysis was performed for gene expression signatures of aging and OSKM reprogramming.

## Transcriptomic clock analyses

To assess the transcriptomic age (tAge) of fibroblasts treated with chemical reprogramming reagents, we applied multi-tissue mouse transcriptomic clocks based on the identified signatures of aging (*Tyshkovskiy et al., 2023*). Filtered RNAseq count data was passed to log transformation and scaling. The missing values corresponding to clock genes not detected in the data were imputed with the precalculated average values. Pairwise differences between average tAges of fibroblasts subjected to chemical reprogramming and corresponding controls were assessed using independent t-tests.

## Measurement of splicing damage

The RNA sequencing reads were aligned to the mouse genome (*mm10*) using STAR with the parameters '`--chimSegmentMin 2 --outFilterMismatchNmax 3 --alignEndsType` EndToEnd', and

Percent spliced-in (Psi) values were calculated using rMATS (*Shen et al., 2014*). To identify the alternative splicing changes after compounds treatment, we compared the alternative splicing events between 7c treatment and paired control sample and selected the events with significant differences (with the cutoff of |ΔPsi avg| > 0.1, and FDR < 0.05). To quantify the overall damage caused by splicing changes, we also counted the proportion of the events that may disrupt protein function. In detail, we predicted the functional consequence for each alternative splicing event and counted the following three categories for each sample: (1) Introduction of a premature stop codon; (2) Production of frame-shifted proteins; and (3) Loss of a peptide in a protein domain region.

## Sample processing for quantitative TMT-proteomic analysis

For TMT18-plex proteomics, a slightly modified form of published procedures was utilized (*Li et al., 2021*; *Navarrete-Perea et al., 2018*). In brief, fibroblasts were seeded at a density of 1 million cells per 10 cm dish and were treated with the specified compounds for 6 days (media replaced once after 3 days) at 37 °C, 5% $CO_2$, 3% $O_2$. Following completion of the incubation period, cells were rinsed twice with cold PBS and scraped and collected in 0.4 ml of modified RIPA buffer (50 mM $KPO_4$ (pH 7.5), 150 mM NaCl, 1% Triton X-100, 0.5% deoxycholate, 1% SDS, and 1 X protease and phosphatase inhibitors). Samples were then vortexed for 30 seconds, placed on ice for 10 minutes, and pelleted at 21,000x$g$ for 10 minutes (4 °C). The supernatants were transferred to fresh microfuge tubes, and protein concentration was measured by BCA assay (ThermoFisher, Waltham, MA). Then, 200 μg of each protein sample was reduced with 5 mM TCEP, alkylated with 10 mM iodoacetamide, quenched with 10 mM DTT, and chloroform-methanol precipitated. The protein pellets were briefly dried in a speed-vac and resuspended in 100 μl of 200 mM EPPS, pH 8.5. Protein samples were digested with 1:100 Lys-C:protein overnight at room temperature, followed by 1:100 trypsin:protein for 6 hours at 37 °C.

The samples were then labeled with tandem mass tag (TMTpro) reagents (*Li et al., 2021*). Acetonitrile was added to a final volume of 30% prior to adding the TMTpro labeling reagent. For protein level analysis, ~50 μg of peptides were labeled with 100 μg of TMT. For phosphopeptide analysis, we estimated the phosphopeptide enrichment to be ~1.5:100 and so ~30 μg of peptides were labeled with 60 μg of TMT. Labeling occurred at room temperature for 1 hour. Approximately 2 μg of peptide from each sample was pooled, desalted and analyzed by mass spectrometry to check labeling efficiency.

TMT labeling efficiency was determined using the 'label-check' sample, which is a quality control step prior to the final pooling of TMT-labeled samples. Here, we combined a small amount (1–3 μl, or ~2 μg) of each sample and analyzed it by mass spectrometry to confirm that peptide digestion was successful, if the degree of labeling was sufficient, and if the labeled samples contained approximately equal amount of peptides. During database searching, the TMTpro label was considered a variable modification at the N-terminus and at lysine residues. We then determined the labeling efficiency for the N-terminus and the lysine residues by dividing labeled N-terminal peptides by total peptides and then labeled lysine-containing peptides by the total lysine-containing peptides. The labeling efficiency should be greater than 95% before proceeding with the analysis. Once labeling efficiency was verified (here, > 97%), hydroxylamine was added at a final concentration of ~0.3% and incubated for 15 minutes at room temperature. The remaining samples were pooled at a 1:1 ratio across all channels.

To enrich phosphopeptides, the pooled sample was desalted over a 200 mg SepPak column and phosphopeptides were enriched with the Pierce High-Select Fe-NTA Phosphopeptide enrichment kit following manufacturer's instructions. The eluate was desalted via StageTip (*Rappsilber et al., 2003*) and was ready for MS analysis. The washes and the unbound fraction of this enrichment were desalted and used for proteome-level analysis.

For proteome-level analysis, the pooled sample was fractionated using basic-pH reversed-phase (BPRP) Liquid Chromatography using an Agilent 1260 pump with an Agilent 300 Extend C18 column (2.1 mm ID, 3.5 μm particles, and 250 mm in length). The flow rate over the column was 0.25 ml/min and we used a 50 minute linear gradient with 5–35% acetonitrile in 10 mM ammonium bicarbonate (pH 8). Ninety-six fractions were collected and concatenated into 24 superfractions prior to desalting (*Paulo et al., 2016*). These 24 superfractions were sub-divided into two groups, each consisting of 12 non-adjacent superfractions. These superfractions were subsequently acidified with 1% formic acid and vacuum centrifuged to near dryness. Each superfraction was desalted via StageTip (*Rappsilber*

et al., 2003). Once dried by vacuum centrifugation, the sample was reconstituted using 5% formic acid and 5% acetonitrile prior to acquisition of LC-MS/MS data.

## Mass spectrometry data collection and processing

Mass spectrometric data were collected on an Orbitrap Fusion Lumos mass spectrometer, which was coupled to a Proxeon NanoLC-1200 UHPLC and a FAIMSpro interface (Saba et al., 2009). The 100 µm capillary column was packed with 35 cm of Accucore 150 resin (2.6 µm, 150 Å; Thermo Fisher Scientific).

Data for the protein superfractions were acquired on an Orbitrap Lumos with a CV set of −40/−60/−80 V over a 90 minute gradient using Realtime search (RTS)-MS3. The scan sequence began with an MS1 spectrum (Orbitrap analysis, resolution 60,000, 350–1350 Th, automatic gain control (AGC) target $5\times10^5$, maximum injection time 100 milliseconds). MS2 analysis consisted of collision-induced dissociation (CID), quadrupole ion trap analysis, automatic gain control (AGC) $2\times10^4$, NCE (normalized collision energy) 35, q-value 0.25, maximum injection time 120 milliseconds, and isolation window at 0.5 Th. RTS was enabled and quantitative SPS-MS3 scans (resolution of 50,000; AGC target $2.5\times10^5$; max injection time of 250 milliseconds) with a real-time false discovery rate filter implementing a modified linear discriminant analysis. For FAIMS, the dispersion voltage (DV) was set at 5000 V, the compensation voltages (CVs) used were −40 V, −60 V, and −80 V, and the TopSpeed parameter was set at 1 second.

For phosphopeptide profiling, data were acquired using two injections on an Orbitrap Lumos, one with a CV set of −40/−60/−80 V, and a second injection with a CV set of −30/−40/−50 V, both over a 150 minute gradient. A 1 second TopSpeed cycle was used for each CV. The scan sequence began with an Orbitrap MS1 spectrum with the following parameters: resolution: 60,000, scan range: 350–1350 Th, automatic gain control (AGC): 100%, and maximum injection time: 118 milliseconds. MS2 analysis consisted of higher-energy collisional dissociation (HCD) with the following parameters: resolution: 50,000, AGC: 300%, normalized collision energy (NCE): 36%, maximum injection time: 250 milliseconds, and isolation window: 0.7 Th, and. In addition, unassigned, singly, and >5 + charged species were excluded from MS2 analysis and dynamic exclusion was set to 60 seconds.

Once the spectra were converted to mzXML using MSconvert (Chambers et al., 2012), database searching could be performed. Database searching included all mouse entries from UniProt (downloaded March 2022), which was concatenated with a version of the database in which the order of the amino acid residues of each protein was reversed. Database searches used a 50-ppm precursor ion tolerance and a product ion tolerance of 0.03Da. These wide mass tolerance windows were chosen to maximize sensitivity in conjunction with Comet searches and linear discriminant analysis (Huttlin et al., 2010; Beausoleil et al., 2006). For static modifications, lysine residues and peptide N-termini were modified with +304.207 Da due to the TMTpro labels. Meanwhile, all cysteine residues were modified with iodoacetamide (carbamidomethylation) that results in a+57.021Da increase in mass. Also, methionine oxidation (+15.995Da) was set as a variable modification. Likewise, deamidation (+0.984 Da) at glutamine and asparagine residues and phosphorylation (+79.966 Da) at serine, threonine, and tyrosine residues were also set as variable modifications for phosphopeptide enrichment. The false discovery rate (FDR) was set at 1% at the peptide level with filtering by linear discriminant analysis (Elias and Gygi, 2007). The protein lists were assembled further to a final protein-level FDR of 1%. The intensities of reporter ions were corrected for the isotopic impurities of the different TMT reagents (McAlister et al., 2012). For each protein, the peptide signal-to-noise (S/N) measurements were summed and normalized to account for equal protein loading by equating the sum of the signal for all proteins in each channel.

For further processing, normalized Uniprot ID-level intensities were summed and $\log_2$ transformed to obtain unique intensities at the level of gene symbol identifiers. Differential abundance analysis was performed with msqrob2 (Goeminne et al., 2016; Goeminne et al., 2018a; Goeminne et al., 2018b), where we included treatment (control, 2c, or 7c), age group (4 months or 20 months), and the treatment:age group interaction as covariates. For the phosphoproteomics data, the normalized relative phosphopeptide intensities were $\log_2$ transformed. Differential abundance analysis at the level of these normalized relative $\log_2$ phosphopeptides intensities was again performed in msqrob2 (Goeminne et al., 2018a; Goeminne et al., 2016) with the same covariates as for the proteomics analysis. We performed GSEA to obtain protein-level enrichments from the phosphopeptides. To obtain pathway

enrichments for both the proteomics and phosphoproteomics data, we ran the GSEA algorithm with the HALLMARK, KEGG, and REACTOME ontologies from the Molecular Signature Database (*Subramanian et al., 2005*) (MSigDB). For kinase substrate enrichment analysis, we performed GSEA using the mouse kinase substrate dataset from PhosphoSitePlus (*Hornbeck et al., 2015*) (last updated on April 19th, 2023).

## Seahorse assays

Fibroblasts treated with vehicle, 2c, or 7c for 6 days as described above were seeded on 24-well cell culture plates overnight at 37 °C, 5% $CO_2$, 3% $O_2$ at a cell density of 50,000 cells per well. The following morning, the cells were aspirated and the media was replaced with 500 µl of XF DMEM medium (pH 7.4) supplemented with 1 mM sodium pyruvate, 2 mM glutamine, and 10 mM glucose. Cells were incubated for 1 hour at 37 °C in a non-$CO_2$ incubator prior to assessment of oxygen consumption rate (OCR) and extracellular acidification rate (ECAR) on a XeF24 Extracellular Flux Analyzer (Agilent Technologies, Santa Clara, CA) using a standard Mito Stress protocol (*Divakaruni et al., 2014*). Immediately following the end of the run, cells were stained with LCS1 nuclear green and cells were counted on a SpectraMax i3 plate reader (Molecular Devices, San Jose, CA). OCR and ECAR measurements were then normalized to cell count per well.

## Preparation of metabolite samples

Cells were incubated with their respective treatments for 6 days, and then the cells were rinsed with ice-cold 0.9% saline solution. After rinsing, cells were scrapped into 1 ml of 0.9% saline solution, spun at 2000xg for 2 minutes (4 °C), aspirated, and flash frozen in liquid nitrogen. To extract metabolites, the cells were thawed, lysed in 400 µl of ice-cold methanol, 200 µl of ice-cold chloroform was added, and then another 200 µl of chloroform and 300 µl of ultrapure water was added and the resulting mixture was vortexed for 30 seconds. Following incubation on ice for 10 minutes, the samples were spun at 4000 RPM for 10 minutes (4 °C), and the upper polar phases (600 µl) were transferred to fresh microfuge tubes and stored at –80 °C until ready for analysis by mass spectrometry. For normalization of metabolite peak area values, 400 µl of methanol was added to the leftover samples (consisting of nonpolar phases and precipitated protein), and the samples were vortexed and spun at 20,000xg for 5 minutes (4 °C). Protein pellets were then aspirated, air-dried for 30 minutes, and resuspended in 500 µl of modified RIPA buffer. Protein concentrations were measured by BCA assay (Thermo Fisher, Waltham, MA), and metabolite peak areas were divided by the total amount (µg) of protein in each sample.

## Metabolite profiling

Hydrophobic interaction liquid chromatography (HILIC) positive ionization profiling was carried out on a Shimadzu Nexera X2 U-HPLC (Shimadzu Corp.; Marlborough, MA) coupled to an Exactive Plus hybrid quadrupole Orbitrap mass spectrometer (Thermo Fisher Scientific; Waltham, MA) according to a published procedure (*Rojas-Tapias et al., 2022*). Protein was precipitated from 10 µl of sample by addition of nine volumes of 74.9:24.9:0.2 v/v/v acetonitrile/methanol/formic acid containing internal standards (valine-d8, Sigma-Aldrich; St. Louis, MO; and phenylalanine-d8, Cambridge Isotope Laboratories; Andover, MA). Precipitated material was cleared by centrifugation and the supernatant was injected directly onto a 150x2 mm 3 µm Atlantis HILIC column (Waters; Milford, MA). Elution was as follows: (1) 5% mobile phase A (10 mM ammonium formate and 0.1% formic acid in water), 0.5 minutes, 250 µl/minute; (2) linear gradient to 40% mobile phase B (acetonitrile with 0.1% formic acid), 10 minutes, 250 µl/minute. MS analysis was with electrospray ionization (ESI) in positive ion mode with the following parameters: full scan analysis over 70–800 m/z at 70,000 resolution and 3 Hz data acquisition rate; sheath gas 40; sweep gas 2; spray voltage 3.5 kV; capillary temperature 350 °C; S-lens RF 40; heater temperature 300 °C; microscans 1; automatic gain control target 1e6; and maximum ion time 250 milliseconds.

HILIC negative ionization profiling (*Rojas-Tapias et al., 2022*) was carried out on an Shimadzu Nexera X2 U-HPLC (Shimadzu Corp.; Marlborough, MA) coupled to a Q Exactive Plus hybrid quadrupole orbitrapmass spectrometer (Thermo Fisher Scientific; Waltham, MA). Protein was precipitated from 30 µl of sample by addition of four volumes of 80% methanol containing internal standards (inosine-15N4, thymine-d4 and glycocholate-d4; Cambridge Isotope Laboratories; Andover, MA).

Precipitated material was cleared by centrifugation and the supernatant was injected directly onto a 150x2.0 mm Luna NH2 column (Phenomenex; Torrance, CA). Elution was as follows: (1) 10% mobile phase A (20 mM ammonium acetate and 20 mM ammonium hydroxide in water) and 90% mobile phase B (10 mM ammonium hydroxide in 75:25 v/v acetonitrile/methanol), 400 µl/minute; (2) linear gradient to 100% mobile phase A, 10 minutes, 400 µl/minute. MS analysis was with electrospray ionization (ESI) in the negative ion mode with the following parameters: full scan analysis over m/z 70–750 at 70,000 resolution and 3 Hz data acquisition rate; sheath gas 55; sweep gas 10; spray voltage −3.0 kV; capillary temperature 350 °C; S-lens RF 50; heater temperature 325 °C; microscans 1; automatic gain control target 1e6; and maximum ion time 250 milliseconds.

Raw data was processed using TraceFinder (Thermo Fisher Scientific; Waltham, MA) and Progenesis QI (Nonlinear Dynamics; Newcastle upon Tyne, UK) for Q Exactive (Plus) experiments or MultiQuant (SCIEX; Framingham, MA) for 5500 QTRAP experiments. Metabolite identities were confirmed using authentic reference standards or reference samples.

## Annexin V assay

Cells were seeded in six-well plates at a density of 200,000 cells per well, and treated for either 1, 3, or 6 days with 7c or vehicle (media and treatments replaced after 3 days). Following their specified treatment times, cells were trypsinized, neutralized, and spun at 500x$g$ for 5 minutes at room temperature. Following aspiration, cells were resuspended in 100 µl of Annexin V binding buffer (10 mM HEPES, 2.5 mM CaCl$_2$, 140 mM NaCl) and stained at room temperature with 1 µg/ml DAPI and 5 µl of Annexin V FITC (Tonbo Biosciences, San Diego, CA) for 15 minutes in the dark. Following this incubation period, 400 µl of binding buffer was added to each solution of cells, the cells were filtered through mesh cell strainers, and cells were immediately analyzed by flow cytometry on a Cytek DxP11 (Cytek Biosciences, Fremont, CA). For each sample, 100,000 events were recorded, and cells were gated based on fluorescence signals of unlabeled cells and single-labeled controls in FlowJo (Ashland, OR).

## Prkaca colocalization

Fibroblasts were seeded at a density of 40,000 cells per well in #1.5 chambered coverglass pre-coated with 0.2% gelatin. Cells were then incubated overnight prior to fixation and permeabilization with 4% paraformaldehyde and ice-cold methanol, respectively. Fixed cells were blocked with 3% BSA and stained with Alexa Fluor 488 goat anti-mouse for mouse anti-Prkaca, and Alexa Fluor 568 goat anti-rabbit for rabbit anti-Tom20 (Bio-Techne, Minneapolis, MN), and 10 µg/ml DAPI. Imaging was performed on a Zeiss LSM980 Airyscan2 Point Scanning Confocal using a 63 X/1.4 oil objective. Image analysis was performed in ImageJ (*Schneider et al., 2012*). Training and access to this instrument was provided by the Microscopy Resources on the North Quad (MicRoN) core at Harvard Medical School.

## Prkaca RNAi and measurement of mitochondrial membrane potential

For RNAi, cells were seeded at a density of 100,000 cells per well in six-well plates and incubated overnight at 37 °C, 5% CO$_2$, 3% O$_2$. Then, cells were transfected with either 15 nM Silencer predesigned siRNA targeting Prkaca or a nontargeting siRNA (negative control) using Lipofectamine RNAiMAX Transfection Reagent prepared in Opti-MEM reduced serum media. Cells were cultured for 6 days total, with media and treatments (siRNA:lipofectamine complexes and drug cocktails) replaced after 3 days. After 6 days, membrane potential staining and imaging was performed as previously described.

To verify successful knockdown of Prkaca, cells were trypsinized and pelleted after 3 days of RNAi. Cell pellets were then aspirated, resuspended in SDS-PAGE sample buffer, and resolved on 4–20% TGX gels. Blots were performed using monoclonal antibodies against Prkaca (Santa Cruz Biotechnology Inc, Dallas, TX) and GAPDH (Thermo Fisher, Waltham, MA). Percent knockdown of Prkaca was assessed relative to GAPDH abundance using ImageJ (*Schneider et al., 2012*).

## Statistical analyses and rigor

Each independent biological replicate for this study consists of fibroblasts isolated from a single mouse treated with each cocktail. A technical replicate represents fibroblasts from the same mouse with the same treatment but measured independently. All measurements in this study consist of at least 3 independent biological replicates. The type I error rate was controlled at the 5% level, either by student's

t-test, or by controlling the family-wise error rate at 5% by one-way ANOVA and Tukey's post-hoc analysis, where appropriate. For large omics experiments, the false discovery rate was controlled with the Benjamini-Hochberg procedure at the 5% level, unless stated otherwise.

## Acknowledgements

The authors thank Dr. William Oldham for assistance with Seahorse measurements, Bobby Brooke for coordinating measurements of DNAm by microarray, Jesse Poganik for assistance with analysis of metabolomics data, and members of the Gladyshev lab for helpful discussions. The project described was supported by the National Institute of Biomedical Imaging and Bioengineering, National Institutes of Health through grant number T32 EB016652 to WM, by NIA grants to VNG, and by NIH/NIGMS grant R01 GM132129 (to JAP, SPG) and GM67945 (to SPG).

## Additional information

### Funding

| Funder | Grant reference number | Author |
| --- | --- | --- |
| National Institute on Aging | R01AG067782 | Vadim N Gladyshev |
| National Institute of Biomedical Imaging and Bioengineering | T32EB016652 | Wayne Mitchell |
| National Institute of General Medical Sciences | R01GM132129 | Joao A Paulo Steven P Gygi |
| National Institute of General Medical Sciences | R01GM67945 | Steven P Gygi |

The funders had no role in study design, data collection and interpretation, or the decision to submit the work for publication.

### Author contributions

Wayne Mitchell, Conceptualization, Data curation, Formal analysis, Validation, Investigation, Visualization, Methodology, Writing - original draft, Writing - review and editing; Ludger JE Goeminne, Alexander Tyshkovskiy, Sirui Zhang, Julie Y Chen, Formal analysis, Validation, Investigation, Visualization, Methodology, Writing - review and editing; Joao A Paulo, Data curation, Investigation, Methodology, Writing - review and editing; Kerry A Pierce, Angelina H Choy, Data curation, Validation, Investigation, Methodology, Writing - review and editing; Clary B Clish, Steven P Gygi, Supervision, Investigation, Methodology, Project administration; Vadim N Gladyshev, Conceptualization, Supervision, Funding acquisition, Validation, Project administration, Writing - review and editing

### Author ORCIDs

Wayne Mitchell http://orcid.org/0000-0003-0871-2080
Sirui Zhang http://orcid.org/0000-0002-1992-3345
Steven P Gygi http://orcid.org/0000-0001-7626-0034
Vadim N Gladyshev http://orcid.org/0000-0002-0372-7016

### Ethics

All experiments using mice were performed in accordance with institutional guidelines for the use of laboratory animals and were approved by the Brigham and Women's Hospital and Harvard Medical School Institutional Animal Care and Use Committees under Protocol # 2016N000368.

Reviewer #1 (Public Review): https://doi.org/10.7554/eLife.90579.3.sa1
Reviewer #2 (Public Review): https://doi.org/10.7554/eLife.90579.3.sa2
Author Response https://doi.org/10.7554/eLife.90579.3.sa3

## Additional files

### Supplementary files
• Supplementary file 1. Raw genecounts. Raw genecounts from mRNA-seq experiments obtained following STAR mapping.

• Supplementary file 2. Proteomics and phosphoproteomics datasets. Normalized peptide and phosphopeptide abundances obtained from TMT 18plex experiments.

• Supplementary file 3. Splicing raw data. Observed alternative splicing events.

• MDAR checklist

### Data availability
Sequencing and microarray data have been deposited in GEO under accession code GSE247199. All data generated or analyzed during this study are included in the source data and/or supporting files.

The following dataset was generated:

| Author(s) | Year | Dataset title | Dataset URL | Database and Identifier |
|---|---|---|---|---|
| Mitchell W, Goeminne LJE, Tyshkovskiy A, Zhang S, Chen JY, Paulo JA, Pierce KA, Choy AH, Clish BC, Gygi SP, Gladyshev VN | 2024 | Multi-omics characterization of partial chemical reprogramming reveals evidence of cell rejuvenation | https://www.ncbi.nlm.nih.gov/geo/query/acc.cgi?acc=GSE247199 | NCBI Gene Expression Omnibus, GSE247199 |

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
