## [Editor Report · eLife assessment]

This **important** study reports comprehensive multi-omics data on the changes induced in young and aged male mouse dermal fibroblasts after treatment with chemical reprogramming factors. The authors provide **solid** evidence to support their claim that chemical reprogramming factors induce changes consistent with a reduction of cellular 'biological' age (e.g., correlations with established aging markers in whole tissues).

---

## [Referee Report · Reviewer #1 (Public Review)]

Summary:

The investigators employed multi-omics approach to show the functional impact of partial chemical reprogramming in fibroblasts from young and aged mice.

Strengths:

Multi-omics data was collected, including epigenome, transcriptome, proteome, phosphoproteome, and metabolome. Different analyses were conducted accordingly, including differential expression analysis, gene set enrichment analysis, transcriptomic and epigenetic clock-based analyses. The impact of partial chemical reprogramming on aging was supported by these multi-source results.

---

## [Referee Report · Reviewer #2 (Public Review)]

The short-term administration of reprogramming factors to partially reprogram cells has gained traction in recent years as a potential strategy to reverse aging in cells and organisms. Early studies used Yamanaka factors in transgenic mice to reverse aging phenotypes, but chemical cocktails could present a more feasible approach for in vivo delivery. In this study, Mitchell et al sought to determine the effects that short-term administration of chemical reprogramming cocktails have on biological age and function. To address this question, they treated young and old mouse fibroblasts with chemical reprogramming cocktails and performed transcriptome, proteome, metabolome, and DNA methylation profiling pre- and post-treatment. For each of these datasets, they identified changes associated with treatment, showing downregulation of some previously identified molecular signatures of aging in both young and old cells. From these data, the authors conclude that partial chemical reprogramming can rejuvenate both young and old fibroblasts.

The main strength of this study is the comprehensive profiling of cells pre- and post-treatment with the reprogramming cocktails, which will be a valuable resource for better understanding the molecular changes induced by chemical reprogramming. The authors highlighted consistent changes across the different datasets that are thought to be associated with aging phenotypes, showing reduction of age-associated signatures previously identified in various tissues.

---

## [Author Response]

The following is the authors’ response to the original reviews.

eLife assessmentThis valuable study reports comprehensive multi-omic data on the changes induced in young and aged male mouse tail fibroblasts after treatment with chemical reprogramming factors. The authors claim that chemical reprogramming factors induce changes consistent with a reduction of cellular 'biological' age (e.g., correlations with established aging markers in whole tissues). However, the study relies on previously identified aging markers (instead of aging in the tail fibroblast system itself), and thus, at this stage, the evidence in support of the observed molecular changes truly reflecting changes in biological age in the study system is still incomplete.
**Essential revisions**
After discussion with reviewers, we believe that the conclusions of the manuscript would be significantly strengthened with the following revisions:(1) Rather than basing the analysis of age-related markers on public tissue data, it is recommended that authors use their own data on pre-reprogramming fibroblasts to define molecular aging-related markers/signatures specifically for male tail fibroblasts at 4 vs 20 months. This should also always be included in figures as reference points.

We appreciate these helpful comments. Please refer to our responses to Reviewers #1 and #2 concerning these suggestions and the corresponding changes we have made in the revised manuscript.

(2) In general, the methods as written lack the details necessary to fully understand the study/reproduce it independently, notably in terms of data analysis choices (e.g. use of FWER/FDR type correction for multiple testing, use of raw vs normalized RNA counts for PCA, etc).

Thank you for this feedback. We have modified our text to address this issue. Please refer to our responses to Reviewer #1 for the specific changes we have made.

(3) More generally, the authors should better outline the limitations/caveats of their experimental design in the discussion and/or abstract, including the specific cell type and the choice of using only male data (since aging itself is very sex-dimorphic, and the impact of partial reprogramming on aging phenotypes may also be sex-dimorphic).

Thank you for this important feedback. We have now added a section to our Discussion in which we directly address potential limitations of our study concerning sex-specific differences and the cell type used.

**Public Reviews:**

**Reviewer #1:**
Summary:The investigators employed multi-omics approach to show the functional impact of partial chemical reprogramming in fibroblasts from young and aged mice.Strengths:Multi-omics data was collected, including epigenome, transcriptome, proteome, phosphoproteome, and metabolome. Different analyses were conducted accordingly, including differential expression analysis, gene set enrichment analysis, transcriptomic and epigenetic clock-based analyses. The impact of partial chemical reprogramming on aging was supported by these multi-source results.

We appreciate the reviewer noting the strength and comprehensiveness of our approach.

Weaknesses:More experimental data may be needed to further validate current findings.

We thank the reviewer for this suggestion. To further validate our findings, we have proceeded as follows:(1) First, we have investigated the role of Prkaca activation during partial chemical reprogramming with 7c (see updated Fig. 5C, Fig. 5 – figure supplement 1B). By confocal microscopy, we show that partial chemical reprogramming with 7c does not cause Prkaca to localize to mitochondria; rather, its cellular distribution is altered to favor nuclear localization. We also use RNAi to knockdown Prkaca and find that Prkaca is not necessary for mediating the increase in mitochondrial membrane potential upon partial chemical reprogramming with 7c.

(2) We have determined the effect of partial chemical reprogramming with 7c on apoptosis using Annexin V assay (see updated Fig. 5 – figure supplement 1C). We show that during the course of partial chemical reprogramming, the proportion of apoptotic cells steadily increases to about 20 percent.

(3) We have re-analyzed our multi-omics data to determine the molecular differences (e.g. at the epigenome, transcriptome, proteome, and metabolome levels) between fibroblasts isolated from young and old mice (see updated Fig. 2 – figure supplement 1, Fig. 6 – figure supplement 1, and Fig. 7 – figure supplement 2). Additionally, we have updated Fig. 7A to include statistical comparisons of transcriptomic age of 4-month-old and 20-month-old fibroblasts. Finally, we have updated Fig. 3D to include functional enrichment of gene and protein expression levels of aged fibroblasts.

(4) We have more thoroughly characterized the effects of partial chemical reprogramming on the epigenome (see Fig. 7 – figure supplement 3).

(5) Julie Y. Chen was added on as an additional co-author for producing the analyses shown in Fig. 7 – figure supplement 2, and Fig. 7 – figure supplement 3.

**Reviewer #2:**
The short-term administration of reprogramming factors to partially reprogram cells has gained traction in recent years as a potential strategy to reverse aging in cells and organisms. Early studies used Yamanaka factors in transgenic mice to reverse aging phenotypes, but chemical cocktails could present a more feasible approach for in vivo delivery. In this study, Mitchell et al sought to determine the effects that short-term administration of chemical reprogramming cocktails have on biological age and function. To address this question, they treated young and old mouse fibroblasts with chemical reprogramming cocktails and performed transcriptome, proteome, metabolome, and DNA methylation profiling pre- and post-treatment. For each of these datasets, they identified changes associated with treatment, showing downregulation of some previously identified molecular signatures of aging in both young and old cells. From these data, the authors conclude that partial chemical reprogramming can rejuvenate both young and old fibroblasts.The main strength of this study is the comprehensive profiling of cells pre- and post-treatment with the reprogramming cocktails, which will be a valuable resource for better understanding the molecular changes induced by chemical reprogramming. The authors highlighted consistent changes across the different datasets that are thought to be associated with aging phenotypes, showing reduction of age-associated signatures previously identified in various tissues. However, from the findings, it remains unclear which changes are functionally relevant in the specific fibroblast system being used. Specifically:(1) The 4 month and 20 month mouse fibroblasts are designated "young" vs "old" in this study. An important analysis that was not shown for each of the profiled modalities was a comparison of untreated young vs old fibroblasts to determine age-associated molecular changes in this specific model of aging. Then, rather than using aging signatures defined in other tissues, it would be more appropriate to determine whether the chemical cocktails reverted old fibroblasts to a younger state based on the age-associated changes identified in this comparison.

In our study, we have used 4 biological samples per group for young and old untreated fibroblasts, and these samples have been used to calculate the effect of 7c and 2c cocktails on gene expression in each age group. Therefore, the correlation between logFC induced by 7c/2c treatment and logFC between young and old fibroblasts would be biased, since the same untreated samples would be used in both calculations: estimates B-A and C-B will be, on average, negatively correlated even if A, B and C are independent random variables. For this reason, to investigate the effect of cocktails on biological age, we utilized gene expression signatures of aging, estimated based on more than 2,600 samples of different ages from 25 data sources (PMID: 37269831). Notably, our multi-tissue signatures of aging were identified based on data from 17 tissues, including skin. Therefore, these biomarkers seem to represent more reliable and universal molecular mechanisms of aging. Since they have been identified using independent data, the signatures also don’t introduce the statistical bias described above. For these reasons, we think that they are more applicable for the current analysis. To demonstrate that the utilized aging signatures are overall consistent with the changes observed in studied fibroblasts, we performed GSEA-based analysis, testing association between logFC in aged fibroblasts and various signatures of aging and reprogramming (similar to our analysis in Fig. 2E). We found that the changes in aged fibroblasts from the current study demonstrated positive association with the majority of aging signatures (kidney, liver and multi-tissue signatures in mouse and rat) (Fig. 2 – figure supplement 1A) and were negatively associated with signatures of reprogramming. In addition, we characterized functional changes perturbed in untreated aged fibroblasts at the level of gene expression and protein concentrations and observed multiple changes consistent with the aging signatures, such as upregulation of genes and proteins involved in inflammatory response and interferon signaling (Fig. 3D, Fig. 2 – figure supplement 1C). Therefore, changes observed in untreated aged fibroblasts seem to agree with age-related molecular changes identified across mammalian tissues in our previous studies.

We would also like to mention that the epigenetic clocks used in this study consistently show that the fibroblasts from 20-month-old fibroblasts are significantly older than the fibroblasts from 4-month-old mice (Fig. 7B). Moreover, we have revised the manuscript to show that these epigenetic differences between young and old untreated fibroblasts are not due to overall changes in mean DNA methylation (Fig. 7 – figure supplement 2). In contrast, in the revised manuscript, we observe that 7c treatment is reducing the epigenetic age of cells by decreasing mean DNA methylation levels (Fig. 7 – figure supplement 3).

(2) Across all datasets, it appears that the global profiles of young vs old mouse fibroblasts are fairly similar compared to treated fibroblasts, suggesting that the chemical cocktails are not reverting the fibroblasts to a younger state but instead driving them to a different cell state. Similarly, in most cases where specific age-related processes/genes are being compared across untreated and treated samples, no significant differences are observed between young and old fibroblasts.

We agree that our data shows that partial chemical reprogramming seems to induce a similar effect on young and old fibroblasts. In Fig. 2 – figure supplement 1B, the Spearman correlation coefficients for the effects on gene expression in young and old fibroblasts are 0.80 and 0.85 for 2c and 7c, respectively. It is important to note that the effect of partial chemical reprogramming is a magnitude higher (say in terms of number of differentially expressed genes) than the effect of aging in the untreated fibroblasts. Partial chemical reprogramming with 7c, we believe, is pushing the cells to a younger state as a byproduct of producing a different cellular metabolic state with a strong increase in OXPHOS capacity.

(3) Functional validation experiments to confirm that specific changes observed after partial reprogramming are indeed reducing biological age is limited.

Functional validation of rejuvenating interventions is limited in vitro, as cells do not completely maintain their “aged” phenotype once isolated and cultured, and pursuing partial chemical reprogramming in vivo in naturally-aged mice was beyond the scope of the study. One of the best reporters of biological age that are preserved in primary cells in vitro are epigenetic and transcriptomic clocks, which were both utilized in this manuscript to show that 7c treatment, but not 2c, reduces biological age. We show that splicing-related damage is marginally elevated in old fibroblasts compared to young, and that 7c reduces splicing damage by reducing intron retention. Moreover, the epigenetic clocks used in this study show that the 20-month-old fibroblasts are significantly older than the 4-month-old fibroblasts, indicating that the “aged” phenotype is at least partially preserved. Furthermore, according to previous studies (PMIDs: 37269831, 31353263), one of the strongest functional biomarkers of aging is downregulation of mitochondrial function and energy metabolism, including oxidative phosphorylation, while upregulation of these functions is usually associated with extended lifespan in mice. For this reason, we have focused on these pathways in our study and assessed them with functional assays.

(4) Partial reprogramming appears to substantially reduce biological age of the young (4 month) fibroblasts based on the aging signatures used. It is unclear how this result should be interpreted.

This is a caveat of all reprogramming strategies/”anti-aging” interventions developed and tested to date. Currently, there are no genetic or pharmacological methods that target only the “aged” state and not the “young” state as well (i.e. an intervention that would only cause a change in old cells and revert them to a younger state). However, “young” cells in our study and many other studies are still the cells of an intermediate age, as aging appears to begin early during development. Therefore, perhaps unsurprisingly, partial chemical reprogramming seemed to have similar effects on fibroblasts isolated from young and old mice, which is in line with OSK/OSKM reprogramming. These results should be interpreted as follows: partial chemical reprogramming does not depend on the epigenetic state (biological age) of adult cells to induce rejuvenation. We have updated the discussion section of our manuscript accordingly.

**Recommendations for the authors:**

**Reviewer #1:**
(1) How was the PCA conducted for RNA-seq data? Were the raw or normalized counts used for PCA?

Normalized counts were used for PCA of the RNA-seq data.

(2) Supplementary Fig 3c, why was the correlation between the red rows and red columns low? Was the color of group messed up? Why was the Pearson correlation used instead of Spearman correlation? Most of the correlation analyses in the manuscript used Spearman correlation.

We thank the reviewer for noticing this mistake. The colors of the groups have now been corrected. Furthermore, to be consistent with the rest of the manuscript, we have performed a Spearman correlation analysis on the normalized proteomics data to evaluate sample-to-sample similarities and updated Fig. 3 – figure supplement 1 accordingly. Overall, the results are similar to those obtained by Pearson correlation.

(3) Were the significant metabolites tested by one-way ANOVA adjusted for family-wise type I error rate? It is surprising that over 50% metabolites were significant.

Yes, the significant metabolites were adjusted for family-wise type I error rate (with a 5% significance threshold) in Fig. 6B.

(4) Missing full names of several abbreviations, such as NIA, RLE, PSI, etc.

Thank you for noticing the missing abbreviations. We have corrected this by writing out the full term in the first instance in which each abbreviation appears.

(5) Methods section may be too long. Some paragraphs could be moved to supplementary text.

eLife does not have a limit to the number of figures or amount of text. Therefore, we have kept the methods section largely unaltered as we feel that they would be helpful to the scientific community.

**Reviewer #2:**
(1) As discussed in the public review, I would recommend first establishing what differences exist between 4 month and 20 month fibroblasts to identify potential age-related changes in these fibroblasts.

We thank the reviewer for this suggestion. We have now thoroughly characterized the molecular differences between fibroblasts taken from young and old mice at the epigenome, transcriptome, proteome, and metabolome levels. Please refer to previous responses for more specific details.

We have also attempted to establish aging-related differences at the phosphoproteome level, particularly in regards to mitochondrial processes (see figure below), but only GOcc: mitochondrion and GObp: mitochondrial transport come close to being statistically significant (raw p-values of 0.05 and 0.08, respectively) in the control comparison.

(2) While the global changes currently highlighted in the study are informative and should remain in the revised manuscript, additional analyses to show which age-related changes identified in point 1 are reverted upon 2c or 7c treatment would better address the question of whether these cocktails revert age-related changes seen in fibroblasts. These analyses should be performed for each dataset (i.e transcriptomic, proteomic, epigenomic, metabolomic) generated.

Thank you for this comment. We have now evaluated the effects of partial chemical reprogramming on the specific molecular differences between fibroblasts isolated from young and old mice (see updated Fig. 2 – figure supplement 1, Fig. 6 – figure supplement 1, Fig. 7 – figure supplement 2, and Fig. 7 – figure supplement 3). For functional enrichment of aged fibroblasts at the gene and protein level, please refer to updated Fig. 3D.

(3) Comparisons between partial reprogramming and OSKM reprogramming signatures are repeatedly made in the paper, but it is not clear from the text whether similarity to OSKM reprogramming signatures is a desired or undesired feature. Since there are likely both rejuvenating and oncogenic aspects of the OSKM signatures, it is unclear what conclusions can be made from these comparisons.

Two central questions of this study were (1) if partial chemical reprogramming could induce cellular rejuvenation, and (2) if so, would it do so by merely chemically activating expression of Yamanaka factors. In this study, we find that 7c, the cocktail that demonstrated the most profound effect on biological age, only minorly upregulates Klf4, downregulates c-Myc, and has no effect on Sox2 or Oct4 expression. Thus, partial chemical reprogramming seems to operate through a mechanism independent of upregulating OSK/OSKM gene expression. This is crucial as it suggests that there are other transcription factors outside of OSKM that can be targeted to induce cellular rejuvenation and reversal of biological age. However, the direct transcriptional targets of partial chemical reprogramming are currently unknown and require further investigation.

Partial reprogramming with OSK/OSKM has several limitations, including low efficiency, oncogenic risk, and differences in the speed of reprogramming according to cell/tissue type. These risks could be inherently tied to the transcription factors OSKM themselves; thus, partial chemical reprogramming, by avoiding strong activation of these genes, could potentially avoid these risks and provide a safer means for reversing biological age in vivo. However, extensive follow-up studies beyond the scope of this manuscript are certainly required to determine this.

We have addressed this comment by modifying the discussion to include these points.

(4) When analyzing the phospho-proteomics data, results are discussed as general changes in phosphorylation of proteins involved in different cellular processes. However, phosphorylation can either activate or inhibit a specific protein, and can depend on the specific residue in a protein that is modified. Different proteins in a cellular process can also respond in opposite directions to phosphorylation. Treating activating and inactivating phosphorylation events separately in describing these results would be more informative.

We agree that an analysis that considers for each specific phosphosite whether it activates or inactivates a particular pathway would in principle be preferable over our current enrichment analysis that only accounts for the increase or decrease in phosphorylation of each site without knowing its biological meaning. However, unfortunately, we think it is currently practically not possible to conduct such an analysis. The proposed analysis would require a database with information on which residues are (de-)phosphorylated when a certain pathway is activated. However, as far as we know, there are currently no databases that link activation or inactivation of specific phosphosites to pathways in repositories like KEGG, HALLMARK, GObp, GOcc, GOmf, Reactome, etc.

Some databases link phosphosites to drugs, diseases and kinases (e.g. PTMsigDB (PMID: 30563849)). However, these authors explicitly state: “We note that we do not capture functional annotations of PTM sites in PTMsigDB, such as activating or inactivating effect on the modified protein.” Furthermore, even in these databases, for the vast majority of the registered phosphosites, the responsible kinases are unknown, especially in mice. In our work, we made use of PhosphoSitePlus for kinase substrate enrichment analysis (see Fig. 5B). Such analyses, where kinase activity is inferred based on activated phosphosites are indeed commonly performed (see PMIDs: 34663829, 37269289, 37585503).

In the absence of a repository that assigns activity to phosphosites, if enrichment analysis is being done for biological pathways, it is standard practice to so without accounting for whether phosphosites are activating or inactivating (see PMID: 34663829), as we have done in our manuscript (Fig. 5A).

Despite the drawbacks, we believe our analysis is relevant, as it demonstrates important biological activity in these pathways uopn 2c/7c treatments as compared to controls. For example, the observed increase in abundance in mitochondrial OXPHOS complexes (Fig. 3E) combined with an increase in general phosphorylation of mitochondrial proteins (Fig. 5A) likely points to an increase mitochondrial activity, although one cannot exclude that some individual phosphorylation events might have inhibitory effects on certain mitochondrial proteins, while others might indicate increases in activity.

(5) For the transcriptomic and epigenetic aging clocks used in Fig 7, significance tests need to be included for untreated 4 month vs 20 month fibroblasts. Particularly for the transcriptional clock, the differences are small and suggest that it may not be a strong aging signature.

We have updated our clock analysis with the most recent versions of the clocks and added statistical significance between 4-month-old and 20-month-old untreated fibroblasts there (Fig. 7A). The difference is statistically significant for the chronological clock. However, when the lifespan-adjusted clock was applied, no statistical significance was observed, suggesting that 20-month-old fibroblasts do not exhibit substantial changes in gene expression associated with decreased healthspan and increased mortality.

(6) For heatmaps shown in Figure 3D and Figure 4, please include untreated 4 month and 20 month fibroblasts as well to determine if pathways being compared are different between young and old fibroblasts.

We have updated Figure 3D with functional enrichment results for aged fibroblasts at gene and protein expression levels, as requested. As for Fig. 4, we explained in our reply to point 1 of Reviewer #2 in the public review why addition of aged fibroblasts there would be biased there. Instead, we have performed GSEA-based association analysis for changes observed in aged fibroblasts and signatures of aging (Fig. 2 – figure supplement 1), confirming that our signatures are overall consistent with patterns of 20-month-old fibroblasts from the current study.